# PERTURBED EXAMPLES REVEAL INVARIANCES SHARED BY LANGUAGE MODELS

## ABSTRACT

An explosion of work in language is leading to ever-increasing numbers of available natural language processing models, with little understanding of how new models compare to better-understood models. One major reason for this difficulty is saturating benchmark datasets, which may not reflect well differences in model performance in the wild. In this work, we propose a novel framework for comparing two natural language processing models by revealing their shared invariance to interpretable input perturbations that are designed to target a specific linguistic capability (e.g., Synonym-Invariance, Typo-Invariance). Via experiments on models from within the same and across different architecture families, this framework offers a number of insights about how changes in models (e.g., distillation, increase in size, amount of pre-training) affect multiple well-defined linguistic capabilities. Furthermore, we also demonstrate how our framework can enable evaluation of the invariances shared between models that are available as commercial black-box APIs (e.g., InstructGPT family) and models that are relatively better understood (e.g., GPT-2). Across several experiments, we observe that large language models share many of the invariances encoded by models of various sizes, whereas the invariances encoded by large language models are only shared by other large models. Possessing a wide variety of invariances may be a key reason for the recent successes of large language models, and our framework can shed light on the types of invariances that are retained by or emerge in new models.

## 1 INTRODUCTION

A key reason for the tremendous progress and adoption of natural language processing (NLP) models has been the ready availability of models that can be effectively adapted to diverse downstream tasks and datasets (Wolf et al., 2019). However, with the increasing number of new models, it is difficult to know how new models compare to better-understood ones. This is complicated by the fact that standard benchmark datasets are saturating (Dehghani et al., 2021; Owen, 2023), and small differences on these datasets may in fact correspond to large differences in model performance in the wild (Tay et al., 2022; Zhang et al., 2022; Liu et al., 2023).

To enable more comprehensive model comparisons, we propose a novel framework for comparing two natural language processing models by investigating their shared invariance to specific input perturbations. We focus specifically on evaluating invariances that are shared between models, as the invariances encoded by a model implicitly define the features of data that the model deems important and is consequently sensitive to, as well as delineate the features it finds irrelevant. Understanding the shared invariances of two NLP models can reveal finer-grained similarities and differences of the instance properties utilized by the two models for their predictions (Shah et al., 2023).

While evaluating shared invariance is important, not all invariances are created equal: some invariances may be desirable (e.g., invariance to synonym substitution for machine translation systems) while others may be undesirable (e.g., invariance to word order of image captioning systems). Therefore, we enable the evaluation of specific shared invariances via interpretable input perturbations that are designed to target a specific linguistic capability (e.g., Synonym-Invariance, Typo-Invariance). A linguistic capability evaluates a model's competence on a particular aspect of knowledge and understanding required to solve an NLP task by validating its input-output behavior under the corresponding scenario. For instance, the linguistic capability 'Synonym-Invariance' evaluates whether

a sentiment analysis model changes its prediction if the positive verb is replaced by its synonym. Hence, the generated perturbations along specific linguistic capabilities enable us to measure shared model invariances along different linguistic capabilities.

We demonstrate the utility of our proposed framework in deriving novel insights about how changes in models such as distillation and increase/decrease in size affect shared invariances along multiple well-defined linguistic capabilities. We also show how our framework can be used to compare how invariances along different linguistic capabilities evolve over the course of pre-training for a particular model. Additionally, we also demonstrate how our framework can enable evaluation of the invariances shared between models that are available as commercial black-box APIs (e.g., InstructGPT family) and models that are relatively better understood (e.g., GPT-2). Across several experiments, we find that while larger language models share many of the invariances encoded by models of varying scale, invariances encoded by large language models are only shared by other large models of similar sizes.

Our main contributions can be summarized as follows: (1) We propose a novel framework for defining linguistic capabilities w.r.t a reference NLP model to generate interpretable invariant perturbations. (2) We propose two novel measures: Hard-SCoPE and Soft-SCoPE to measure the degree of shared (behavioral) invariances between two models along a particular linguistic capability. Both measures evaluate the extent to which a target model behaviorally aligns with the reference model along linguistic capabilities. (3) Through experiments on two NLP tasks—text classification and language modeling—we uncover several insights, such as: distilling BERT leads to loss of shared-invariances along certain linguistic capabilities (such as Typo-Invariance) more than others (Synonym-Invariance); models (within an architecture family) tend to have a higher (or similar) degree of shared-invariances with models of larger sizes compared to other models of similar sizes. Moreover, we also evaluate the shared invariances between *task-specific* finetuned models and *generalist* instruction-tuned models (from the InstructGPT family) that are available as black-box APIs.

## 2 RELATED WORKS

**Representations:** Numerous works have proposed methods for analyzing and comparing representations of NLP models (Morcos et al., 2018; Saphra & Lopez, 2019; Liu et al., 2019; Durrani et al., 2021). Most notably, Wu et al. (2020) investigate the representational similarity of NLP models at multiple levels (i.e., at both neuron and layer-level output) to quantify the effects of different design choices across models from both across and within architectural families. Along similar lines, Phang et al. (2021) explore the effects of fine-tuning a neural language encoder by comparing representations of a fine-tuned language encoder with its pre-trained counterpart across layers. Most similar to our work, Nanda et al. (2022) proposed a novel measure (STIR) to quantify the similarity between representations of two models via measuring their shared invariances. They achieve this by first generating a set of perturbations that don't change the representations of one model and consequently measuring the extent to which the other model's representations are invariant on them. However, this setup is not directly applicable to NLP due to the discrete nature of language input, where representation inversion would lead to perturbations along arbitrary directions in the input space and consequently linguistically inconsistent samples (La Malfa & Kwiatkowska, 2022). We address this by generating invariant perturbations (for a particular model) along well-characterized and interpretable linguistic capabilities by using discrete optimization techniques. Finally, while a central theme of our work is also comparing the similarities and differences between two NLP models, we present an orthogonal approach that focuses on behavior instead of representations.

**Behavior:** Many works compare the behavioral similarity between two models (trained for a given task) by evaluating the difference between their average performances on the held-out "test-set" (e.g., IID accuracy, perplexity, etc). For example, previous work has used IID (Independent and identically distributed) accuracy to evaluate the effect of well-defined design choices such as model architecture and training scheme (Ding et al., 2021), training time constraints (Geiping & Goldstein, 2022), and latency and memory (Sanh et al., 2019). However, recently many researchers have highlighted the limitations of IID test-sets in identifying different failure modes (Hooker et al., 2019; 2020) and have consequently proposed alternative approaches for rigorous evaluation (Rychalska et al., 2019; Prabhakaran et al., 2019; Ribeiro et al., 2020; Ribeiro & Lundberg, 2022). Most relevant to our work, Ribeiro et al. (2020) proposed CheckList–a methodology for evaluating the behavior of

NLP models along general linguistic capabilities that are applicable for most NLP tasks. More recently, La Malfa & Kwiatkowska (2022) defined linguistic capabilities as symbolic perturbations of an input sentence for a particular task, and evaluated whether a model's predictions for this sentence align with human annotators. While the above approaches can highlight differences between the two models' ability to generalize under the perturbations introduced by a linguistic capability, they perform an indirect behavioral comparison via the human annotators. In this work, we provide a complementary approach that directly evaluates shared behavioral invariances between two models by defining linguistic capabilities with respect to an NLP model instead of a human annotator.

## 3 METHODOLOGY

One way to evaluate a model's linguistic capabilities is via perturbations i.e., perturbing the input and evaluating whether the model's prediction on the perturbed input aligns with human judgment. However, in contrast to computer vision, where perturbed inputs can be optimized directly in an end-to-end manner, constructing perturbations for discrete language inputs involves deciding on many different factors (e.g., type of perturbation, constraints on perturbation, etc). Hence in this work, we leverage key insights from the NLP adversarial robustness literature to decompose perturbations corresponding to a linguistic capability $C$ in terms of four independent components: *transformation*, *constraints*, *goal function* and *search method*. Thus, perturbations along each capability (e.g., Synonym-Invariance) attempt to manipulate an input text ('I love watching movies.') to produce its perturbed counterpart ('I love watching films.') using well-defined transformations (WordNet synonym swapping) and constraints (disallowing stopword modifications) to satisfy the goal function (i.e., reference model $m$ remains behaviorally invariant).

### 3.1 GOAL FUNCTION AND SEARCH METHOD

To effectively quantify measures such as shared invariances (defined in Sec. 3.3) between a reference and a target NLP model, we enforce that the reference model is invariant to the perturbation introduced by the linguistic capability. The behavioral invariance serves as the *goal function* while generating perturbations with respect to the *reference NLP model*, ensuring that the perturbation generation process interacts with the reference NLP model. This formulation is important as invariance-based measures are otherwise difficult to measure using purely observational data (Nanda et al., 2022). Additionally, this also lends directionality to our shared-invariance measures as the perturbations generated w.r.t two different reference models would be different, allowing us to delineate invariances unique to any model and measure their degree of overlap with others.

We define the goal of behavioral invariance at the level of the output softmax probabilities i.e., the reference model $m$ is behaviorally invariant if there is a negligible difference between the predicted probability distribution on the base and perturbed sample. More formally, consider an NLP model $m$ that outputs probability distribution $m(x)$ for an input text $x$. A linguistic capability $C$ perturbs $x \in X$ s.t. $m$ is invariant to the perturbed text $x'$

$$C(x; m) = \underset{x'}{\operatorname{argmin}} \mathcal{L}(m(x), m(x')), \text{ subject to } x \neq x' \tag{1}$$

where $\mathcal{L}$ is an objective function that guides the optimization process. Since $x$ is a sequence of tokens, we use discrete optimization techniques (e.g., greedy search) for finding $x'$ that minimizes $\mathcal{L}(m(x), m(x'))$ in the finitely large transformation space. The optimization techniques can vary in computational costs depending on the linguistic capability. For instance, we require $\approx 2.5 \times$ time for generating perturbations for Synonym-Invariance compared to Typo-Invariance (refer supplementary Sec. K for details). In our experiments, we define $\mathcal{L}(m(x), m(x')) = \|m(x) - m(x')\|_1$. In practice, we observe that minimizing this objective leads to $x'$ that are at least invariant in argmax predictions. We provide more details in the supplementary Sec. E. Note that the objective function can take other forms as long as it captures differences in both direction and magnitude between $m(x)$ and $m(x')$. We present results for the same in supplementary Sec. I.

### 3.2 TRANSFORMATIONS AND CONSTRAINTS

Once the goal function and search method are specified, we can fully formalize different linguistic capabilities by specifying the corresponding transformations and constraints. In this work, we

primarily focus on two such linguistic capabilities: *Synonym-Invariance* and *Typo-Invariance* that perform perturbations at multiple levels (i.e., character-level transformations to word-level substitutions). **Synonym-Invariance** – Synonym-Invariance perturbs words in the input text by replacing them with their synonyms. More specifically, we adopt the transformation strategy proposed by Ren et al. (2019) that determines candidate synonyms for a particular word based on WordNet (e.g., A man laughs out loud. → A man laughs out loudly.). **Typo-Invariance** – Typo-Invariance perturbs a word in the input text by swapping their middle characters (i.e., all characters in a word except the first and last one). Thus, while Synonym-Invariance perturbs input text at a word level, Typo-Invariance produces transformations at a character level (e.g., A man laughs out loud. → A man laughs out luod.). For both linguistic capabilities, we use a greedy search-based method to traverse through candidate transformations and constrain modifications of words that are stopwords, have lengths less than four, or are already perturbed. We focus on these two capabilities because there is a rich literature studying them, albeit from an adversarial robustness perspective as they concern the reliability of many real-world systems, such as spam detection, toxicity classification, and fake news detection (Lee & Ng, 2005; Pruthi et al., 2019; Ren et al., 2019). We perform experiments along an additional linguistic capability: Fairness and report our insights in the supplementary Sec. C due to space constraints. We would like to emphasize that the list of linguistic capabilities (e.g., negation, word order) can be easily expanded by defining the specific transformation and constraints.

## 3.3 METRICS FOR QUANTIFYING BEHAVIORAL-SIMILARITY

We perform experiments with a number of popular metrics (such as accuracy and agreement-rates) as well as propose novel ones (behavioral shared invariances). These metrics can be broadly categorized into three classes: performance-based (Gap in IID accuracy), agreement-based (IID and OOD (Out-of-Distribution) agreement), and invariance-based (Hard-SCoPE, Soft-SCoPE). The invariance-based metrics offer a complementary lens on the behavioral similarity between two NLP models as we empirically observe that the existing metrics often fail to adequately capture them for many kinds of models one would want to investigate. While some of these metrics have been used in the computer-vision literature, their role in determining behavioral similarity has been underexplored in NLP. Hence, we discuss all of them in detail below.

**Notation:** Consider a task $T$ with an IID test set denoted as $(X_{\text{test}}, y_{\text{test}}) \sim \text{D}_{\text{test}}$. Models $\mathcal{M} = \{m_1, m_2\}$ are fine-tuned on training samples for this task, where each model $m$ maps an input $x$ to output a probability distribution $m(x) \in \mathbb{R}^n$ over $n$ unique labels/vocabulary for $T$. The model's prediction $y_m(x)$ is defined as: $y_m(x) = \underset{k \in [n]}{\text{argmax}}\ m(x)_k$ where $m(x)_k$ denotes the probability score for $k^{th}$ label. We aim to assess the behavioral similarity between $m_1$ and $m_2$ along a particular linguistic capability $C$. Perturbations are applied to a set of base samples – $X$ (typically $X_{\text{test}}$).

### 3.3.1 PERFORMANCE-BASED METRICS

Comparing the gap between aggregate performance-based measures is one of the most common ways to characterize behavioral similarity between two models as models with lower performance gaps are generally thought of as more behaviorally similar (Ding et al., 2021; Klabunde et al., 2023). Since we are primarily investigating text-classification tasks in the main paper, we choose accuracy as the performance measure and consequently evaluate the gap between accuracies of two models on the test-set. We present results on language modeling task in the supplementary Sec.- G.

The Gap in IID accuracy is simply the absolute difference in the accuracies of the reference and target models, i.e., **Gap in IID Accuracy:** $|acc(m_1) - acc(m_2)|$, where accuracy of a model $m$ is defined as $acc(m) = \mathbb{E}_{x,y \sim \text{D}_{\text{test}}} \mathbb{1}[y = y_m(x)]$.

### 3.3.2 AGREEMENT-BASED METRICS

Instead of focusing on the average differences in performance, the agreement rates (between $m_1$ & $m_2$) explore the behavioral similarity directly at the instance level. We calculate model agreements on both base and perturbed samples.

$$\textbf{IID Agreement:} \mathbb{E}_{x \in X} \mathbb{1}[y_{m_1}(x) = y_{m_2}(x)]$$

$$\textbf{OOD Agreement:} \mathbb{E}_{x \in X} \mathbb{1}[y_{m_1}(C(x; m_1)) = y_{m_2}(C(x; m_1))]$$

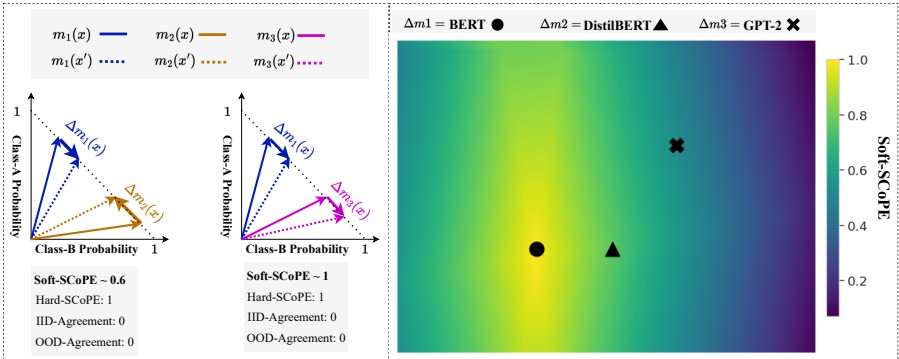

Figure 1: *Left:* Probability vectors for three models ($m_1$, $m_2$, and $m_3$) trained on a binary classification task. For perturbation $x \to x'$, where $x' = C(x; m_1)$, both $m_2$ and $m_3$ satisfy the Hard-SCoPE criteria. However, the effect of the perturbation is more aligned for $m_1$ & $m_3$ (blue and purple) compared to $m_1$ & $m_2$ (blue and brown). *Right:* 2-D Soft-SCoPE surface for a pair of base and perturbed samples, where the reference model BERT is compared with DistilBERT and GPT-2.

Agreement rates between two models is akin to comparing the similarity between their decision regions, especially for out-of-distribution (OOD) data, which represents points sampled along linguistic capabilities in the data manifold (Somepalli et al., 2022).

### 3.3.3 PROPOSED INVARIANCE-BASED METRICS

While agreement rates quantify whether two models have similar behavior on a particular sample, shared invariances can reveal finer-grained similarities of the instance properties utilized by the two models for their predictions. For instance, if a model becomes *variant* to perturbations (e.g., synonym word substitution) that it was originally *invariant* to after a particular design choice (e.g., distillation), it indicates heightened sensitivity towards perturbed aspects of data. Agreement rates are ill-equipped for such investigations as they compare the similarity between predictions of two models. However, measuring shared invariances requires evaluating the effect of a perturbation within each model individually and subsequently comparing if both models exhibit invariance. Thus, measuring behavioral shared invariances amounts to quantifying the extent to which a target model is invariant on perturbations that do not cause any change in the reference model's behavior (refer Eq. 1). Since behavior itself can be measured at different granularity i.e., with respect to exact class prediction or predicted softmax probabilities, we propose two novel notions (Hard and Soft) of measuring behavioral shared-invariance: **SH**ared-**C**apabilities-thr**O**ugh-**P**erturbed-**E**xamples (**SCoPE**)

**Hard-SCoPE:** We want to measure to what degree target model $m_2$'s prediction remains invariant to a change in the input i.e., $x \to x'$, where $x' = C(x; m_1)$, for which the reference model $m_1$ was invariant. We label this quantity as Hard-SCoPE as it considers the 'hard' argmax predictions to determine behavioral shared-invariances.

$$\text{Hard-SCoPE}(m_2 \mid m_1) = \mathbb{E}_{x \in X} \mathbb{1}[y_{m_2}(x) = y_{m_2}(C(x; m_1)) \mid y_{m_1}(x) = y_{m_1}(C(x; m_1))]. \quad (2)$$

Note that the Hard-SCoPE is not calculated between two models (like agreement-rates), but rather between the base and perturbed samples for a particular target model $m_2$. For a binary-classification setup, Hard-SCoPE can be seen as "agreement between agreement-rates" i.e., Hard-SCoPE would be 1 if either both IID and OOD agreement are 0 or both are 1 for a particular base-perturbed sample pair. We discuss the relationship between IID-, OOD-agreements and Hard-SCoPE in more detail in the supplementary Sec. E.

**Soft-SCoPE:** A softer (and perhaps more informative) notion of shared-invariances is to look beyond argmax predictions and investigate whether the perturbation in input space produces the same effect (change) in the output probability distributions of both models. The effect of the perturbation $x \to x'$, where $x' = C(x; m_{ref})$ is generated with reference model $m_{ref}$, in the output probability

distributions for a model $m$ is denoted by $\Delta\vec{m}$ i.e., $\Delta\vec{m} = m(x') - m(x)$. We present an intuition of when a softer notion of shared invariance can be useful in Fig. 1 (left).

In Fig. 1 (left), we visualize the predicted probability distributions of three models $m_1$, $m_2$, and $m_3$ trained on a binary classification task on both base – $m_1(x)$, $m_2(x)$, $m_3(x)$ and perturbed samples – $m_1(x')$, $m_2(x')$, $m_3(x')$, where the perturbation $x'$ is generated by a linguistic capability $C$ and reference model $m_1$ i.e., $x' = C(x; m_1)$. While the input perturbation qualifies as a (hard) shared invariance for both $(m_2 \mid m_1)$ and $(m_3 \mid m_1)$ since both $m_2$ and $m_3$ remain invariant in their argmax predictions; the effect (i.e., change in the predicted output probability distribution) of the perturbation is much more aligned for one pair (i.e., $\Delta\vec{m_3}$ and $\Delta\vec{m_1}$) than the other (i.e., $\Delta\vec{m_2}$ and $\Delta\vec{m_1}$). Thus, the reliance on (argmax) predictions to quantify shared-invariances by Hard-SCoPE could obscure key differences between the effect of the perturbation.

As motivated by this example, a desiderata for the soft shared-invariance metric is to obtain high-values if the change in both models ($\Delta\vec{m_1}$, $\Delta\vec{m_2}$) is similar (in both direction and magnitude) and low-values otherwise. Thus, we define Soft-SCoPE($m_2 \mid m_1$) as:

$$\mathbb{E}_{x \in X}\text{decay}(\text{dist}(\Delta\vec{m_1}, \Delta\vec{m_2}))\mathbb{1}[y_{m_2}(x) = y_{m_2}(C(x; m_1)) \mid y_{m_1}(x) = y_{m_1}(C(x; m_1))], \quad (3)$$

where $\text{decay}(\text{dist}(\Delta\vec{m_1}, \Delta\vec{m_2}))$ is an additional term that weighs the contribution of each pair of base and perturbed samples by a function of the corresponding changes in model probabilities $\Delta\vec{m_1}$ and $\Delta\vec{m_2}$. More specifically, the function is composed by two functions: a function $\text{dist}$ that computes the difference between the changes in model probabilities, and a decay function $\text{decay}$, which has a range $[0, 1]$ i.e., $0 \le \text{decay}(\text{dist}(\Delta\vec{m_1}, \Delta\vec{m_2})) \le 1$ and is monotonically decreasing to ensure lower $\text{dist}$ values correspond to higher similarity, as in all the previous metrics. In our experiments, we chose $\text{dist}(\Delta\vec{m_1}, \Delta\vec{m_2}) = \|\Delta\vec{m_1} - \Delta\vec{m_2}\|_1$ and decay as a linear function. In order to visualize the variation in Soft-SCoPE values between different model pairs i.e., $(m_2 \mid m_1)$ and $(m_3 \mid m_1)$ we plot the 2-D plane spanned by vectors $\vec{v_1} = \Delta\vec{m_2} - \Delta\vec{m_1}$ and $\vec{v_2} = \Delta\vec{m_3} - \Delta\vec{m_1}$. Here, $\Delta\vec{m_1}$ corresponds to BERT (reference model), and $\Delta\vec{m_2}$ & $\Delta\vec{m_3}$ refer to DistilBERT and GPT-2 respectively. We note that unlike Hard-SCoPE of two models that can only take binary values i.e., either 0 or 1 for a particular base-perturbed sample pair, Soft-SCoPE varies smoothly. We also observe that models from the same architectural family (BERT and DistilBERT) have higher Soft-SCoPE compared to models from different architectural families (BERT and GPT-2). We expand on the insight in supplementary Sec. F.

### 3.4 CORRELATION BETWEEN METRICS

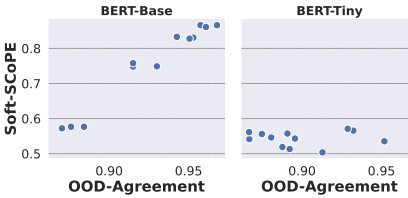

Figure 2: Correlation between proposed invariance-based metrics (Soft-SCoPE) and existing metrics (OOD-Agreement) for different reference and target model pairs. Existing metrics poorly correlate with invariance-based metrics as the size of the reference model is reduced.

Here, we explore whether the proposed invariance-based measures are tightly coupled with metrics previously explored in the literature such as agreement rates. We evaluate the Pearson correlation between OOD-agreement and Soft-SCoPE for different reference and target model pairs from the BERT architecture family and plot the results in Fig. 2. Each column consists of results for different target-reference model pairs for a particular reference model – BERT-Base or BERT-Tiny. We experiment with multiple different target models varying in size with BERT-Base being the largest and BERT-Tiny the smallest. We find that OOD-agreement and Soft-SCoPE are poorly correlated when using comparatively smaller models as reference models i.e., r=0.011 for BERT-Tiny, whereas they are positively correlated when using relatively larger models as reference models i.e., r=0.97 for BERT-Base. Thus, the invariances shared between two NLP models are not necessarily explained by existing metrics. Importantly, the finding that existing metrics are especially poor at capturing shared invariance when the reference model is smaller than the target model further highlights the need for separately evaluating shared-invariances as smaller models are more amenable to controlled analysis (such as circuit analysis (Wang et al., 2022)) and hence likely to be used as reference models. We present results for correlations with other metrics such as IID-agreement and Gap in IID accuracy in the supplementary Sec. J.

## 4 Effect of Model Design Choices on Shared-Invariances

In this section, we investigate the effect of different design choices on the invariances shared by two models. Thus, we experiment with a range of NLP models varying in training (BERT Devlin et al. (2019), DistilBERT (Sanh et al., 2019)), and size (BERT-Tiny, Mini, Small, Medium, Base). Unless otherwise stated, we finetune all architectures for 5 epochs on Stanford Sentiment Treebank (SST2 a binary sentiment classification dataset) (Socher et al., 2013). We present additional results for different datasets (AG-News) and tasks (language modeling) in the supplementary Sec. B and Sec. G. We build upon the "textattack" library (Morris et al., 2020) to implement several linguistic capabilities based on our requirements. For each capability and reference-model pair we generate the perturbed examples three times and report the average results with standard errors.

### 4.1 Different Linguistic Capabilities

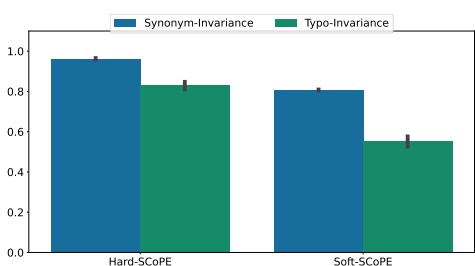

Figure 3: [Reference Model: **BERT**, Target Model: **DistilBERT**]. Comparing shared-invariances between DistilBERT and BERT on Synonym-Invariance and Typo-Invariance defined w.r.t BERT. Distillation hurts some capabilities (Typo-Invariance) substantially more than others (Synonym-Invariance).

In this section, we aim to investigate whether a design choice (i.e., distillation) that has a nominal impact on the IID accuracy, preserves shared invariances along different linguistic capabilities. Specifically, we fix BERT as the reference model & DistilBERT as the target model and compare shared capabilities along Synonym-Invariance and Typo-Invariance.

**Gap in IID accuracy may overestimate the degree of shared invariances:** Both BERT and DistilBERT achieve high accuracies on the SST2 test-set i.e., 93% and 89.45% respectively (3.55% Gap in IID accuracy). However, in Fig 3, we note that a low gap in generalization performance on the IID test-set doesn't necessarily ensure high shared invariances. For instance, DistilBERT is substantially less aligned to BERT along Typo-Invariance. Thus, Gap in IID accuracy may overestimate the degree of shared invariances between two models along a linguistic capability.

**Distilling BERT affects some linguistic capabilities more than others:** In Fig. 3, we also observe that DistilBERT is significantly more similar to BERT along Synonym-Invariance compared to Typo-Invariance. Thus not only is there a decrease in shared-invariances after distillation, but distillation also affects different linguistic capabilities to varying degrees. We posit that this trend can be attributed to the Masked Language Modelling (MLM) pre-training procedure that is common to both BERT and DistilBERT. The MLM objective optimizes the model to predict masked words in a sentence correctly, it's plausible that during pre-training a model develops invariances to diverse in-context word-substitutions. Since, both DistilBERT and BERT are pre-trained on the same corpus (i.e., concatenation of English Wikipedia and Toronto Book Corpus, Sanh et al. (2019)), it's highly likely that the learnt word-invariances are shared between them. Similarly, the lower values for Typo-Invariance may be explained by the lack of misspelled words in the training corpus.

### 4.2 Role of Inductive Biases

So far we have discussed the effect of a particular design choice (i.e., distillation) on shared-invariances along different linguistic capabilities. In this section, we explore the effect of changes in architectural inductive biases on a model's behavior along a linguistic capability i.e., Synonym-Invariance. Specifically, we investigate the role of increasing/decreasing the depth and width of hidden-layers on the shared-invariances. To control for potential confounders we finetune a wide array of models (released by Turc et al. (2019)) belonging to the same architecture family (BERT) varying significantly in both number (L) and size (H) of the hidden layers. Specifically, we inves-

Figure 4: [Linguistic-Capability: **Synonym-Invariance**] Analyzing the effect of size on shared-invariances within the BERT architecture family. We observe that OOD-agreement is higher for target models in similar size ranges as the reference model. However, shared-invariances are higher for target models of larger size irrespective of the reference model.

tigate BERT-Tiny (L=2, H=128), BERT-Mini (L=4, H=256), BERT-Small (L=4, H=512), BERT-Medium (L=8, H=512), and BERT-Base (L=12, H=768).

**Different trends across different metrics:** In Fig. 4-a, with BERT-Tiny (smallest model in our investigation) as the reference model, we observe that the OOD agreement-rate indicate that models in similar in size to BERT-Tiny (i.e., BERT-Mini, BERT-Small) have higher similarity with BERT-Tiny than other larger models (i.e., BERT-Medium, BERT-Base). In contrast, the shared-invariances metrics don't follow the same trend as Hard-SCoPE values tend to increase for larger model sizes, and there isn't a substantial difference between the Soft-SCoPE values across different target models.

**Larger models share more invariances with models of any size:** In Fig. 4-b, we repeat the same experiment with the largest model in our investigation–BERT-Base–as the reference model. Surprisingly, we observe that all metrics indicate that models become less similar to BERT-Base as their size decreases. This is in contrast to previous results with BERT-Tiny (smallest model) as the reference model where larger models had poor OOD-agreement and higher (or similar) shared-invariances. Thus, we hypothesize that even though larger models don't agree with the behavior of smaller models from an agreement perspective, they still share the invariances generated by smaller models. Interestingly, the opposite is not true i.e., smaller models don't necessarily share invariances generated w.r.t larger models as well as other larger models. To understand this more generally, we report the average results for all models (as target models) by marginalizing them over all the different reference models. We expect that metrics depending on model size (i.e., agreement rates) should have uniform values across different target models. In contrast, metrics that are tied to larger model sizes (i.e., shared-invariances) should peak for larger models even after averaging. We report the result in Fig. 4-c, which are consistent with our proposition. It's especially interesting that larger models are able to share a more diverse set of invariances (both from other larger and smaller models) even when they are pretrained/finetuned on the same corpus as the smaller models.

## 5 RELATIONSHIP OF FAMILIAR MODELS WITH BLACK BOX APIS

In the previous sections we discussed *specialist* models that are tuned to perform well on a specific task (e.g., sentiment analysis on SST2). However, in recent years the NLP community has shifted focus towards building more *generalist* models that can perform a diverse set of tasks when prompted with appropriate instructions and exemplars. Importantly, these models don't need to update their weights on downstream task-data as the instructions and exemplars can be directly provided *in-context* during inference (Brown et al., 2020; Wei et al., 2022). However, the state-of-the-art of these models are primarily available in the form of black-box APIs released by industrial research labs, with little information available about their training. We explore how one can quantify the behavioral similarity between models released via black-box APIs (InstructGPT family) and models that are widely adopted in practice (e.g. GPT-2). We follow the methodology of Cheng et al. (2023) for estimating output probability distribution over the task-labels (positive and negative sentiment) from InstructGPT models. Specifically, we sample the output multiple times for each input and take the mode as the final prediction and its frequency as the probability score for that particular label.

Firstly, we explore the effect of the size of the target model on the behavioral similarity along Synonym-Invariance defined w.r.t GPT-2 and report the results in Fig. 5 (left). We use InstructGPT

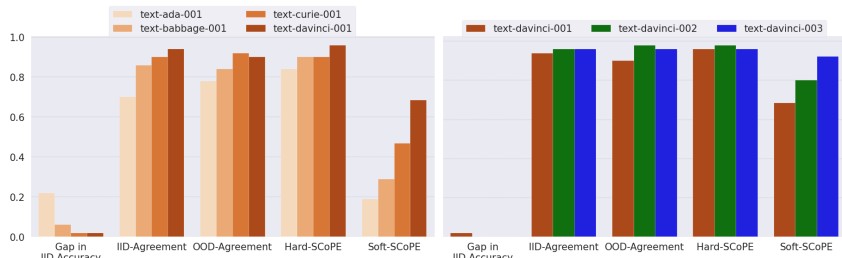

Figure 5: [Reference Model: **GPT-2**, Capability: **Synonym-Invariance**]. Comparing shared-invariances between GPT-2 and various OpenAI models differing in size and finetuning along Synonym-Invariance. Larger InstructGPT models (even with same IID accuracy) share more invariances with GPT-2. Also, state-of-the-art models finetuned with reinforcement learning (text-davinci-003) share more invariances than their supervised finetuned counterparts (text-davinci-002).

models text-ada-001, text-babbage-001, text-curie-001, text-davinci-001 that roughly correspond to model sizes: 350M, 1.3B, 6.7B, and 175B respectively (Gao et al., 2021). We perform all experiments in a zero-shot manner. To reduce costs, we perform experiments on 100 randomly selected samples from the SST2 test-set. The finetuned GPT-2 achieves 96% accuracy on this subset. It cost us ≈ 55$ to compute all the results for this section.

**Larger InstructGPT models share more invariances with GPT-2:** We note that there's a substantial difference in IID performance between the smaller models (text-ada-001) and larger models (text-curie-001, text-davinci-001). Moreover, text-ada-001 is not only less agreeable to the GPT-2 model, but also seems to be substantially less invariant to perturbations that GPT-2 is invariant on i.e., low Hard-SCoPE and Soft-SCoPE. Interestingly, even though text-curie-001 and text-davinci-001 achieve similar IID accuracy (i.e., 94%) there's substantial differences in their shared-invariances. Thus, even though both models seem equivalent based on IID performance, using text-davinci-001 would ensure higher behavioral similarity from the perspective of shared-invariances. Also, this result ties back to our previous observations in Sec. 4.2 about larger models sharing more invariances.

**RL based finetuning may retain more invariances:** We explore the effect of different finetuning methods for instruction following on shared-invariances in Fig. 5 (right). For this, we perform experiments on text-davinci-001, text-davinci-002, and text-davinci-003 models released by OpenAI. text-davinci-001 is finetuned using supervised learning on human and selected model written demonstrations. While, text-davinci-002 utilizes the same objective function, it's pretrained on a mix of text and code. text-davinci-003 differs from text-davinci-002 by using reinforcement learning for finetuning instead of supervised learning. We note that both text-davinci-002 and text-davinci-003 have similar performances across IID accuracy, agreement rates, and Hard-SCoPE. Interestingly, there's a substantial gap in their Soft-SCoPE values, indicating that even though both models remain invariant on an equivalent number of samples (similar hard-scope), text-davinci-003's output probability distribution is more invariant to perturbations generated along Synonym-Invariance.

## 6  CONCLUSION

We propose a framework for evaluating interpretable shared-invariances between two NLP models by evaluating the degree to which a target model shares behavioral similarity on a linguistic capability defined with respect to a reference model. We conduct extensive experiments to highlight the implications of different design choices (e.g. distillation) and find that shared-invariances tend to be affected more along certain linguistic capabilities than others. Furthermore, we also analyze models of different sizes and find that larger target models in general tend to share more invariances. Lastly, we demonstrate the use of our framework in analyzing relationships between black-box APIs and familiar models. One limitation of our current work is inefficient search methods as they need many queries to generate perturbations. Efficient search methods are necessary for generating perturbed samples with reference to a black box APIs. Additionally, while we adopt the approach of Cheng et al. (2023) for estimating predicted probabilities for instruction-tuned models, it would be interesting to evaluate whether our insights vary across different probability estimation methods.

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

# *Appendix for:*
# "Perturbed examples reveal invariances shared by language models"

## A    ADDITIONAL IMPLEMENTATION DETAILS

In this section, we provide an overview of our implementation details. For all our experiments, we use two NVIDIA A40 GPUs with 48GB of memory each. We use the standard model implementations provided by the huggingface transformers library (Wolf et al., 2019). We finetune all models for 5 epochs with a batch size of 64 using Adam optimizer and a linear-drop learning rate schedule with initial value of 2e-5.

## B    ADDITIONAL DATASET: AG'S NEWS

In this section, we present results on an additional dataset – AG's news topic classification dataset (Zhang et al., 2015). It's a multi-class text classification task, where the goal is to classify text from an article into one of four categories i.e., World, Sports, Business, and Sci / Tech. It contains $120,000$ training samples and $7,600$ test samples. Due to compute and time constraints, we randomly sample a subset of $2,000$ samples from the test-set and conduct our experiments on them as base samples (instead of the full test-set). We train models using the same hyper-parameters (learning rate, epochs, etc) as SST2 on the full training set. We repeat the experiments from the main paper and plot the results in Fig. 6 and Fig. 7. We note that the results are qualitatively similar across both the datasets.

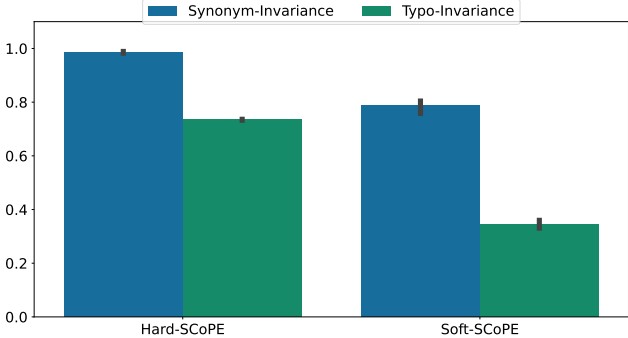

Figure 6: [Dataset: **AG's News**, Reference Model: **BERT**, Target Model: **DistilBERT**]. Comparing shared-invariances between Distil-BERT and BERT on Synonym-Invariance and Typo-Invariance defined w.r.t BERT trained on AG's news dataset. Similar to our observations for SST2 in the main paper, we observe that distillation hurts some capabilities (Typo-Invariance) substantially more than others (Synonym-Invariance).

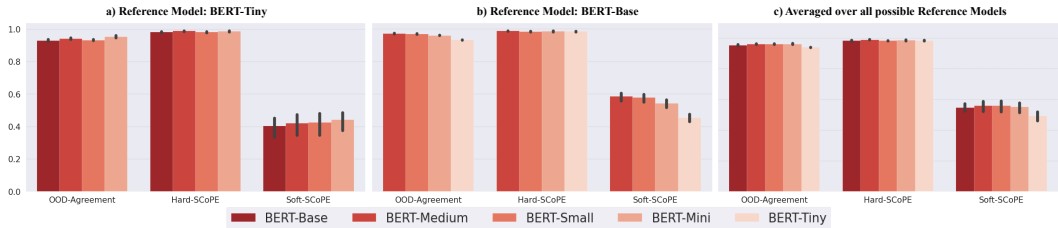

Figure 7: [Dataset: **AG's News**, Linguistic-Capability: **Synonym-Invariance**] Analyzing the effect of size on shared-invariances within the BERT architecture family. Similar to results on the SST2 dataset in the main paper, we observe that larger target models tend to share higher invariances irrespective of the reference model.

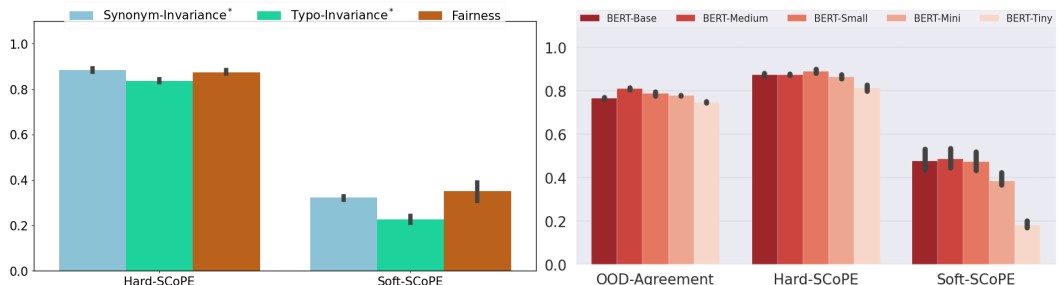

Figure 8: *Left*: [Reference Model: **BERT**, Target Model: **DistilBERT**]. Comparing shared-invariances between DistilBERT and BERT on Synonym-Invariance*, Typo-Invariance*, and Fairness defined w.r.t BERT. While there isn't a substantial difference between shared-invariances along Synonym-Invariance* and Fairness, Typo-Invariance* is lower than both. *Right*: [Linguistic-Capability: **Fairness**] Analyzing the effect of size on shared-invariances within the BERT architecture family. Similar to results on Synonym-Invariance in the main paper, we observe that larger target models share more invariances irrespective of the reference model. Whereas OOD-agreement is higher for models in similar size ranges.

## C    ADDITIONAL LINGUISTIC CAPABILITY: FAIRNESS

In the main paper, we performed experiments along two linguistic capabilities i.e., Synonym-Invariance and Typo-Invariance. In this section, we explore an additional linguistic capability i.e., Fairness. Fairness perturbs the input text ("Men love sports.") by substituting words corresponding to protected categories (such as men) with protected categories (e.g. 'women' ≈ "Women love sports.") from within the same stereotype domain (i.e. Gender). We use a greedy search approach for efficiently finding suitable transformations. We do not adopt any additional constraints on this linguistic capability.

Synonym-Invariance and Typo-Invariance are agnostic to the domain of base samples $x \in X$ i.e., they can be evaluated on any arbitrary set of base samples. In contrast, Fairness is only well defined if $x$ contains words corresponding to different protected categories. Thus, we use the corpus released by Sotnikova et al. (2021) containing sentences with words corresponding to 71 protected categories from 6 different stereotype domains as base-samples $X$ for experiments pertaining to evaluation of Fairness capability. Note, previous work on evaluating linguistic capabilities for a particular model (Ribeiro et al., 2020) also perform a change in base samples (i.e., use samples not necessarily from the test-split) for evaluating certain capabilities in order to decouple *testing* from *implementation*. Additionally, We control for the change in base samples ($X$) by conducting additional experiments on previously studied capabilities, such as Synonym-Invariance and Typo-Invariance, using the new set of base samples. We label them Synonym-Invariance* and Typo-Invariance* respectively. This allows us to draw meaningful comparisons across different capabilities.

In Fig. 8 (left), we first investigate the differences between different linguistic capabilities for a particular design choice. Thus, similar to Sec. 4.1 in the main paper, we fix BERT as the reference model and DistilBERT as the target model. We observe that while Fairness has lower OOD-agreement rate compared to Synonym-Invariance*, there isn't a substantial difference between the shared-invariances (Hard-SCoPE & Soft-SCoPE). Thus, even though DistilBERT disagrees in its predictions with BERT for Fairness more (compared to Synonym-Invariance*), DistilBERT is invariant in its behavior on perturbations generated along Fairness to a similar degree as Synonym-Invariance. Additionally, we also note that Typo-Invariance* shares invariances to a lower degree compared to both Synonym-Invariance* and Fairness further highlighting the role of MLM based training objective as word-substitution is a common perturbation in both Fairness and Synonym-Invariance*. In Fig. 8 (right), we report the shared-invariances between models across different sizes belonging to the same architecture family. Specifically, for each target model we report the averaged results over all possible reference models. Similar to our observations in Sec. 4.2 and Sec. 5 in the main paper for Synonym-Invariance, we observe that larger target models seem to share more invariances (with models of any size) on perturbations generated along Fairness.

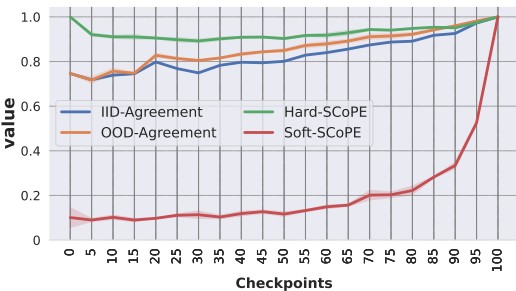

Figure 9: [Reference Model: **final-checkpoint i.e. 100% pre-training**, Target Model: $m_t$ for $t\%$ **pre-training**, Linguistic-Capability: **Synonym-Invariance**] Comparing different metrics to analyze how intermediate-checkpoints share capabilities on Synonym-Invariance defined w.r.t the final-checkpoint. Even though the initial-checkpoint (0% pre-training) is not much better than a random-baseline, it shares a high-degree of Hard-SCoPE and IID/OOD-Agreements. Whereas Soft-SCoPE grows in a gradual manner over the course of pre-training.

## D  PRETRAINING DYNAMICS

In the main paper we focused on evaluating shared capabilities between two models differing in design choices along different linguistic capabilities. Additionally, we can also utilize our framework to empirically understand the dynamics of these linguistic capabilities over the course of pre-training of a language model. This line of analysis can help us probe questions such as: Which linguistic capabilities are *learnt earlier* during pre-training?, How does a linguistic capability *evolve* over the course of pretraining?, etc. To probe such questions effectively, we utilize 21 (equally-spaced) intermediate pre-training checkpoints for BERT released by Sellam et al. (2022). Since, we are primarily interested in quantifying the effect of pre-training (up to a particular checkpoint) in capturing different linguistic capabilities, we refrain from finetuning the full model (on SST2) and rather only train the linear probe layer on top of the frozen base network.

**The curious case of 0% pretraining:**    In Fig. 9, we report values for all the checkpoints evaluated along Synonym-Invariance capability defined with the final-checkpoint (i.e., 100% pretraining) as the reference model (e.g. Soft-SCoPE ($m_t \mid m_{100\%}$ for $t^{th}$ timestamp). We note that the model corresponding to 0% pre-training (finetuned using random initialization) behaves as a random baseline $\approx 0.5$ (or $50\%$) IID accuracy. We find that even though the network is akin to a random baseline, during prediction it outputs only one label: 'positive' (in contrast to predicting both classes with equal probabilities) irrespective of the input. Surprisingly, this model has high IID and OOD agreement rates ($\approx 0.8$ or $80\%$) and the highest possible Hard-SCoPE value i.e. $1$ with respect to the final-checkpoint. On a deeper look, we find that the prediction distribution for the final distribution is also biased towards the 'positive' label. In contrast, the Soft-SCoPE measure increases in a monotonically sublinear fashion over the course of pre-training, indicating that even though the predictions in both IID and OOD states might be similar (high IID/OOD-Agreement rates) and invariant (high Hard-SCoPE) the change in output probability vectors between the IID and OOD predictions varies significantly for the 0% and 100% (final) checkpoints. These observations further reinforce the importance of evaluating a wide range of metrics to gain a holistic understanding of the behavioral similarities between models as certain metrics can be especially deceptive in the low-accuracy regime due to larger possible variance in the underlying model structure.

**Invariances for some capabilities are acquired earlier than others:**    Next in Fig. 10 we look at differences in evolution of different linguistic-capabilities over the course of pre-training. Firstly, in Fig. 10 (left), we observe that Soft-SCoPE (shared-invariances) along Synonym-Invariance is significantly higher compared to Typo-Invariance for the major chunk of pre-training. Note, both of them converge to 1 at 100% pretraining as the Soft-SCoPE of a model with itself is 1 (irrespective of the linguistic-capability). Similar to our observations in Sec. 4.1, we posit that the shared pre-training objective (i.e. MLM) and training corpus leads to a higher degree of shared invariances much earlier in the pre-training for Synonym-Invariance compared to Typo-Invariance, which remains stagnant during most of the pretraining, with a sudden increase towards the end.

**Retaining previously accquired invariances:**    Till now our discussions have revolved around analyzing shared-invariances across different metrics (for a particular capability) and different capabilities (for a particular metric) with the final-checkpoint (i.e. 100% pre-training) as the reference-model. Thus, the central question for previous experiments has been: How (behaviorally) similar

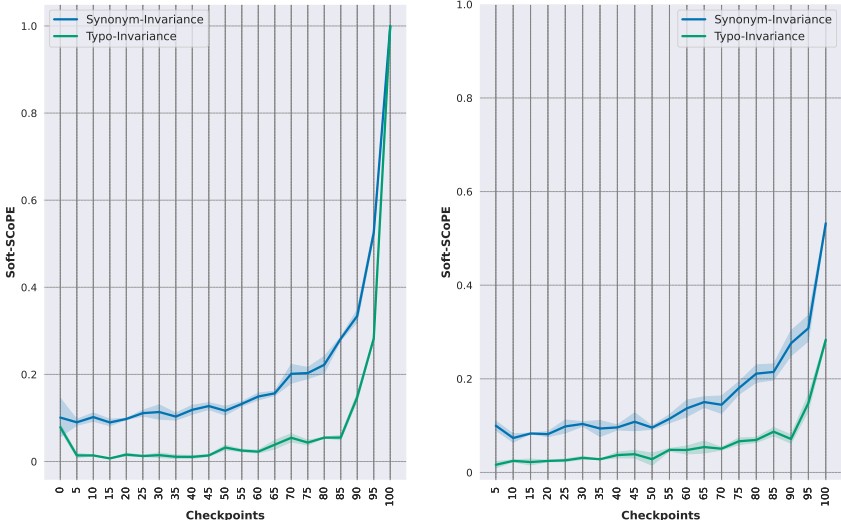

Figure 10: *Left:* [Reference Model: **final-checkpoint i.e. 100% pre-training**, Target Model: $m_t$ **for** $t\%$ **pre-training**, Metric: **Soft-SCoPE**] Comparing the evolution of soft shared-invarinces (i.e. Soft-SCoPE) for different linguistic-capabilities (i.e. Synonym-Invariance, Typo-Invariance) during pre-training. While all trends grow at a monotonically sub-linear pace, invariances for some are acquired earlier than others. *Right* [Reference Model: $m_{t-1}$ **for** $t\%$ **pre-training**, Target Model: $m_t$ **for** $t\%$ **pre-training**, **Soft-SCoPE**] Investigating the retention of previously acquired shared-invariances for different linguistic-capabilities (i.e. Synonym-Invariance, Typo-Invariance) during pre-training. Mild values indicate that many invariances are not retained during the first-half of pre-training, whereas checkpoints become more similar to the final-checkpoint as well as their previous counterparts during the end of pre-training.

is an *intermediate checkpoint to the final checkpoint?* (w.r.t a particular metric along a particular linguistic capability). However, this setup provides little insight regarding whether the models retain their behavioral similarity w.r.t their previous counterparts as well (e.g. Soft-SCoPE ($m_t \mid m_{t-1}$ for $t\%$ pre-training). Thus to probe questions such as: How (behaviorally) similar is an *intermediate checkpoint to its previous counterpart?*, we calculate values of Soft-SCoPE for each checkpoint with the previous checkpoint as the reference-model and report the results in Fig. 10 (right). A low value for a particular checkpoint indicates that the model has changed a lot w.r.t its predecessor while a high value would indicate that the model has retained the previously acquired behavioral invariances. We observe that the Soft-SCoPE values remain centered around mild values for most of the pre-training, indicating that while models are becoming more similar to the final-checkpoint they are only retaining a minor extent of their previously acquired behavioral shared invariances. We note that the shared invariances show a linear increase only towards the very end of the pre-training.

## E    ADDITIONAL EXPLANATION FOR SHARED-INVARIANCES

### E.1    GENERATING INVARIANT PERTURBATIONS

In this section, we analyze properties of perturbations generated along different linguistic capabilities. While the primary goal of generated perturbations is to maintain behavioral invariance with respect to the reference model, it is possible that the search method is unable to find candidates that fulfill this criterion in the finitely large transformation space. Thus, in order to verify whether the generated perturbations are truly behaviorally invariant for the reference model we visualize the distribution of $\mathcal{L}(m(x), m(x'))$ – refer Fig. 11. Note, $\mathcal{L}(m(x), m(x')) = \|m(x) - m(x')\|_1$ is an objective function that penalizes the difference between reference model's output softmax probabilities (behavior) on base and perturbed inputs i.e., lower $\mathcal{L}(m(x), m(x'))$ implies more behavioral invariance (refer Eq. 1).

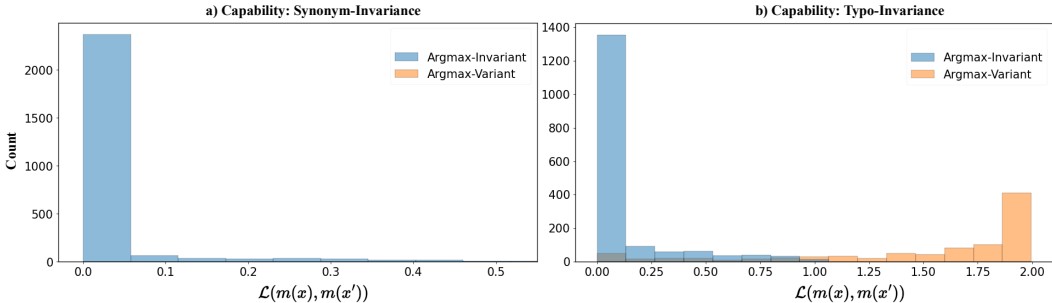

Figure 11: [Reference Model: **BERT**] Distribution of $\mathcal{L}(m(x), m(x'))$ for $x' = C(x; m_1)$ generated along Synonym-Invariance and Typo-Invariance. We note that the distribution is skewed towards lower $\mathcal{L}(m(x), m(x'))$ values and most samples generated are at least invariant in predictions (argmax-invariant).

In Fig. 11 (left) we note that for Synonym-Invariance, the distribution is highly skewed towards lower $\mathcal{L}(m(x), m(x'))$ values indicating that most generated perturbations have minimal difference between the predicted probability distribution on the base and perturbed samples for the reference model and all of them are argmax-invariant i.e., have same prediction on each base-perturbed sample pair. While, for Typo-Invariance (Fig. 11 right), the $\mathcal{L}(m(x), m(x')$ values are higher and there are a few argmax-variant samples as well. Note, the argmax-variant samples would be ignored while evaluating measures such as Hard-SCoPE and Soft-SCoPE (refer Eq. 2 & Eq. 3).

### E.2 Relationship between Agreement-Rates and Shared-Invariances

In this section, we delve deeper into the relationship between agreement-based metrics i.e., IID-agreement & OOD-agreement and invariance-based measures i.e., Hard-SCoPE & Soft-SCoPE. While, Hard-SCoPE doesn't solely depend on any one of IID-agreement or OOD-agreement, looking at both of them together can give indications about the Hard-SCoPE value. For instance, consider a binary classification setup with labels 'Class-A' and 'Class-B' and two models $m_1$ and $m_2$ that have predictions $y_{m_1}(x)$ & $y_{m_1}(x')$ and $y_{m_2}(x)$ & $y_{m_2}(x')$ for a particular base-perturbed sample pair $(x, x')$, where $x' = C(x; m_1)$.

In such a setup, the Hard-SCoPE can be seen as "agreement between agreement-rates" i.e., Hard-SCoPE is 1 only when both agreement-rates are either 0 or both are 1. Hard-SCoPE reaches a value of 1 when $m_2$ has consistent predictions for both IID and OOD inputs ($y_{m_2}(x) = y_{m_2}(x')$), on samples where $m_1$'s predictions are invariant ($y_{m_1}(x) = y_{m_1}(x')$). In a binary classification scenario where only two predictions are possible (Class-A or Class-B), achieving a Hard-SCoPE value of 1 requires either both $m_1$ and $m_2$ to predict the same label, resulting in IID-Agreement and OOD-Agreement both being 1, or they exhibit different predictions, leading to both IID-Agreement and OOD-Agreement being 0 (row 1 and 4 in Tab. 1). In cases where $m_2$ agrees with $m_1$ for IID(OOD) inputs but disagrees on OOD(IID) inputs, the Hard-SCoPE would be 0 as $m_2$ must have changed its prediction after the perturbation, given that $m_1$ is invariant to the perturbation by design (row 2 and 3 in Tab. 1). Importantly, this behavior does not hold for multi-class classification as it's possible for $m_2$ to change its prediction even when both IID and OOD agreement are 0 (row-5 and 6 in Tab. 1).

We also discuss the relationship between Hard-SCoPE and Soft-SCoPE. Soft-SCoPE weighs the contribution of each base-perturbed pair by a function of similarity in their changes in the ouput softmax probability under a perturbation. Importantly, this weight lies between [0, 1]. Thus, Soft-SCoPE is upper-bounded by Hard-SCoPE i.e., $0 \leq$ Soft-SCoPE $\leq$ Hard-SCoPE.

## F Shared-Invariances Across Architecture Families

In this section, we aim to investigate how differences in the architecture family of the reference and target models affect their shared-invariances. Intuitively, one would expect a higher degree of shared-invariances for models having similar architectures, courtesy of common inductive biases

| Setup | $m_1$'s prediction | | $m_2$'s prediction | | IID-Agreement | OOD-Agreement | Hard-SCoPE |
|---|---|---|---|---|---|---|---|
| | $y_{m_1}(x)$ | $y_{m_1}(x')$ | $y_{m_2}(x)$ | $y_{m_2}(x')$ | | | |
| Binary Classification | Class-A | Class-A | Class-B | Class-B | 0 | 0 | 1 |
| | Class-A | Class-A | Class-B | Class-A | 0 | 1 | 0 |
| | Class-A | Class-A | Class-A | Class-B | 1 | 0 | 0 |
| | Class-A | Class-A | Class-A | Class-A | 1 | 1 | 1 |
| Multi-class Classification | Class-A | Class-A | Class-B | Class-B | 0 | 0 | 1 |
| | Class-A | Class-A | Class-B | Class-C | 0 | 0 | 0 |

Table 1: Relationship between IID-agreement & OOD-agreement (agreement-rates) and Hard-SCoPE (shared-invariance). In a binary classification scenario, Hard-SCoPE can be seen as "agreement between agreement-rates" as it's 1 when either IID- and OOD- agreement are both 0 or both 1. However, this relationship doesn't hold for multi-class classification setup.

induced by the architectural family. Thus, to validate this intuition we fix the reference model as BERT and compare shared-invariances of target models both from the same architecture family (DistilBERT) and a different one (GPT-2). We report the results in Fig. 12.

**Models from same architecture family share higher behavioral similarity & invariances:** We observe that the difference between the IID-Accuracies (Gap in IID-Accuracy) is higher for Distil-BERT compared to GPT-2 indicating that when evaluated conventionally, the gap between generalization ability of GPT-2 and BERT would perceived to be smaller than DistilBERT and BERT. However across both linguistic capabilities i.e., Synonym-Invariance and Typo-Invariance, DistillBERT achieves higher IID and OOD agreement rates compared to GPT-2 highlighting when compared at an instance level DistilBERT behaves more similarly to BERT. Interestingly, even though Dis-tilBERT only slightly edges GPT-2 in Hard-SCoPE, there is a substantial difference between their Soft-SCoPE values. This implies that DistilBERT is not only invariant on a large number of samples (that BERT is invariant on), but also the change in the output probability between base and perturbed predictions for DistilBERT is quite similar to that of BERT compared to GPT-2 and BERT.

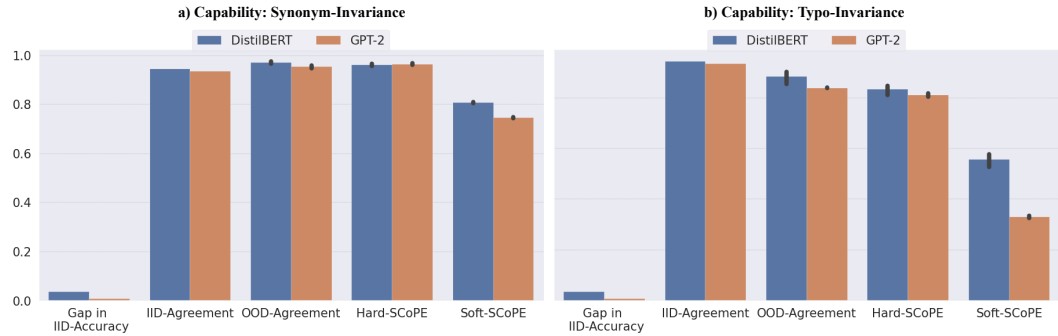

Figure 12: [Reference Model: **BERT**, Linguistic-Capability (left): **Synonym-Invariance**, Linguistic-Capability (right): **Typo-Invariance**] Even though the IID performance gap is smaller between GPT-2 & BERT compared to DistilBERT & BERT. For Synonym-Invariance & Typo-Invariance defined w.r.t BERT, DistilBERT (model from same architecture family) has a higher degree of shared capabilities than GPT-2 (model from different architecture family)

# G ADDITIONAL TASK: LANGUAGE MODELLING

In the main paper, we presented results across multiple linguistic-capabilities for different reference and target model combinations for one particular task i.e., sentiment classification. In this section, we present results on an additional task i.e., language modeling. More specifically, rather than fine-tuning the pre-trained language models on a downstream task and defining a linguistic-capability

w.r.t them, we treat language modeling as a task in itself and define linguistic-capabilities w.r.t the pre-trained language models. We use cosine-similarity for computing agreements as language models have a large vocabulary with many tokens repeating with minor variations. We repeat the experiments presented in the main paper with models from the GPT-2 language modeling architecture family on the SST-2 dataset and plot the results in Fig. 13 and Fig. 14. We note that the results are qualitatively similar to those observed with the BERT model in a classification setup in the main paper highlighting that the effects design choices on linguistic-capabilities investigated in this paper are beyond task-specificities.

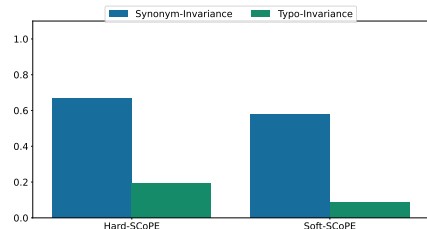

Figure 13: [Dataset: **SST-2**, Reference Model: **GPT-2**, Target Model: **DistilGPT-2**, Task: **Language Modeling**] Similar to results on sentiment-classification, we note that distillation affects shared-invariances across some linguistic-capabilities more than others.

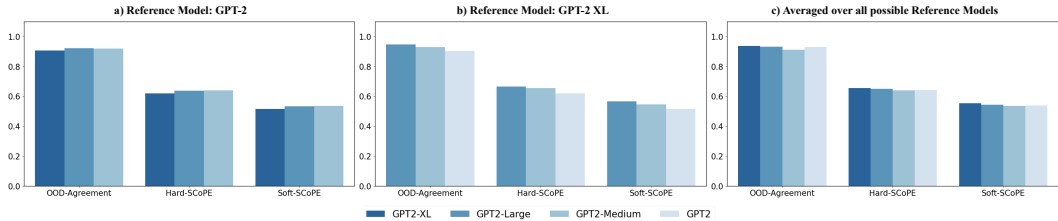

Figure 14: [Dataset: **SST-2**, Linguistic-Capability: **Synonym-Invariance**, Task: **Languag Modeling**] Similar to results on sentiment-classification, we find that larger target models tend to share higher invariances irrespective of the size of the reference model.

# H    SAMPLE COMPLEXITY FOR FRAMEWORK EFFECTIVENESS

In this section, we examine the impact of "number of base samples" on our proposed metrics and report the results in Fig. 15. Specifically, we report the mean metric values and the 95% confidence interval of this estimate computed over 100 trials for many values of base-samples count. We find that our metrics provide tight confidence intervals for as low as 50 base samples. Please note that for the previous experimental results in the main paper and the supplementary we utilize $\approx 1000$ samples.

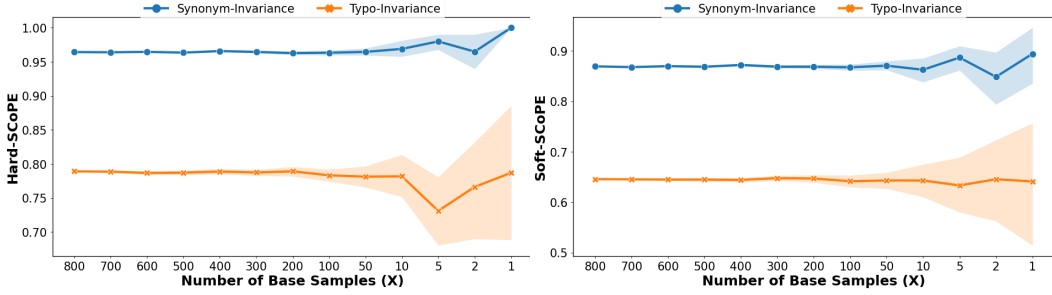

Figure 15: [Reference Model: **BERT**, Target Model: **DistilBERT**]. Examining the impact of base-samples / instances ($X$) count on the proposed metrics i.e., Hard-SCoPE (left) and Soft-SCoPE (right). We report an estimate of the mean metric values and the 95% confidence-interval (y-axis) around this estimate computed over 100 trials for each base-sample count (x-axis). We find that the both the metrics are stable for as low as 50 base-samples with tight confidence-intervals.

## I ADDITIONAL GOAL FUNCTION RESULTS

In the main paper we performed experiments with the L1 norm as the objective function described in Eq. 1. However, it can take other forms as well as long as it captures differences in both direction and magnitude between the reference model's output on base and perturbed samples i.e., $m(x)$ and $m(x')$. In this section, we explore whether our insights are sensitive to the choice of the objective function by employing KL-divergence as the objective function instead. We report the results for one of the analyses in Fig. 16 and observe that there are minimal effects on the overall takeaway when using different objective functions.

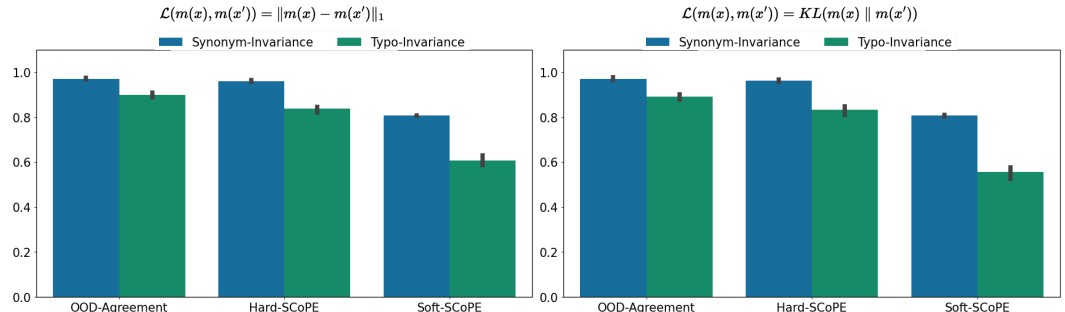

Figure 16: [Reference Model: **BERT**, Target Model: **DistilBERT**]. Analyzing the effect of different objective functions ($\mathcal{L}$) that guide the optimization process of the goal function while generating perturbations for a given reference model. We observe that different objective functions (L1 norm on the left, KL-divergence on the right) have minimal effect on the overall takeaway, i.e., distilling BERT affects some capabilities more than others.

## J ADDITIONAL CORRELATION RESULTS

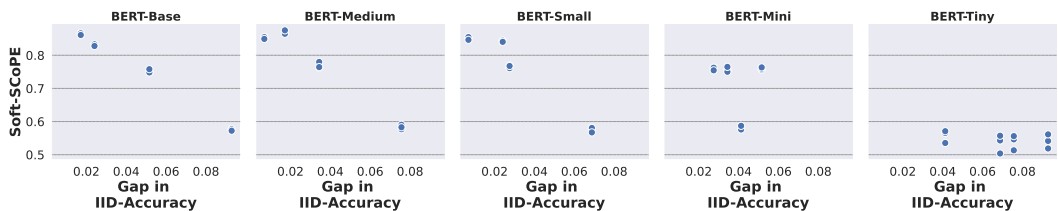

Figure 17: Correlation between proposed invariance-based metrics (Soft-SCoPE) and existing metrics (Gap in IID-Accuracy) for different reference and target model pairs. Similar to results noted int he main paper with OOD-Agreement, Gap in IID-Accuracy also poorly correlate with invariance-based metrics as the size of the reference model is reduced.

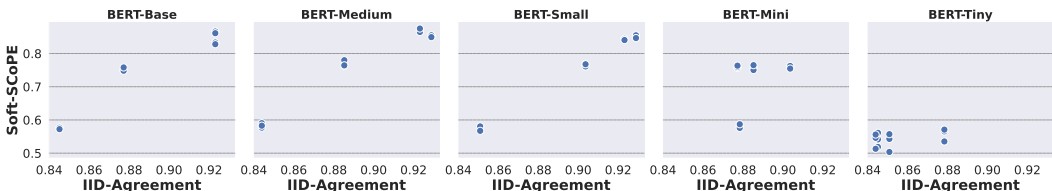

Figure 18: Correlation between proposed invariance-based metrics (Soft-SCoPE) and existing metrics (IID-Agreement) for different reference and target model pairs. Similar to results noted int he main paper with OOD-Agreement, IID-Agreement also poorly correlate with invariance-based metrics as the size of the reference model is reduced.

## K   COMPUTE COSTS

| Linguistic Capability | Time (seconds per sample) |
|---|---|
| Synonym-Invariance | 1.2 |
| Typo-Invariance | 0.48 |

Table 2: [**Reference Model:** BERT-Base, **Dataset:** SST2] Time taken in seconds, required to generate a perturbed sample on a NVIDIA-A100 GPU. The duration varies depending on distinct linguistic capabilities, as certain capabilities are more amenable to the search techniques in generating invariant perturbations than others.

## L   PERTURBATION EXAMPLES

| **Original Sample** | **Synonym-Invariance** | **Typo-Invariance** |
|---|---|---|
| a fast, funny, highly enjoyable movie. | a fast, funny, highly enjoyable film. | a fsat, funny, highly enjoyable movie. |
| my reaction in a word: disappointment. | my response in a word: disappointment. | my reaction in a word: disappointemnt. |
| allows us to hope that nolan is poised to embark a major career as a commercial yet inventive filmmaker. | allows us to trust that nolan is poised to embark a major career as a commercial yet inventive filmmaker. | allows us to hpoe that nolan is poised to embark a major career as a commercial yet inevntive filmmaker. |
| too slow, too long, and too little happens. | too tiresome, too long, and too little happens. | too solw, too long, and too liltte happens. |
| a warm, funny, engaging film. | a warm, comic, engaging film. | a warm, fnuny, engaging film. |

Table 3: [**Reference Model:** BERT-Base, **Dataset:** SST2] Examples of perturbed sentences that are invariant w.r.t the reference model BERT-Base for multiple linguistic capabilities i.e., Synonym-Invariance and Typo-Invariance.

