# OpenReview forum: "Perturbed examples reveal invariances shared by language models"
_ICLR.cc/2024/Conference — Submitted to ICLR 2024_

### Official Review · Reviewer_nq6A · 2023-10-30

**Soundness:** 4 excellent
**Presentation:** 3 good
**Contribution:** 3 good
**Rating:** 6
**Confidence:** 4

**Summary:**

The paper introduces a new framework to measure the difference behavior between two models, with one of them being a reference model and the other one being the target model. Unlike previous work which uses benchmark performance or model prediction agreement, this framework proposes to analyze the shared invariance of the two models when they are given an original input vs. pertubed input which targets specific lingusitic capabilities (i.e., synonym-invariance and typo-invariance).

Experiment results reveal several key findings that (1) the performance gap between models (existing metrics) doesn't always reflect the shared behaviors (invariance) between them, especially where a smaller model is used as a reference model and that (2) larger models tend to share more invariances. The proposed framework closes the first gap by measuring the shared behaviors (invariance) between models.

**Strengths:**

- With the growing popularity of large language models (LLMs) and lots of efforts to make them more efficient, I think the proposed framework would be useful especially for making sure that a particular iteration of a model will have similar or improved behavior compared to the base model. From the analysis, the framework can provide complementary analysis beyond existing metrics.
- Well-executed experiments, with three types of pertubation, and comparison with existing metrics from previous work.
- Application for both specific model (fine-tuned BERT) and more recent generative models (GPT variants).

**Weaknesses:**

- Presentation: some of the details that are important seems to be missing (or rather put in the Appendix). See more details below.

**Questions:**

- Section 3: "four independent components" -> I am not sure if these four are really independent, does transformation and constraint dependent on each other?
- I'm a bit puzzled with the results of BERT-Tiny in Fig. 4, especially regarding the poor OOD-agreement but high invariances. Does this mean that the predictions of both models are quite different (hence poor agreement), but within the individual model the predictions do not change when it is given original and pertubed samples?
- Is there any interesting findings on the model agreement on original input vs. pertubed input?

**Suggestions for presentation improvements**
- Mention the tasks being used in the experiments at the beginning of the paper. It was not clear until Section 3.3 that you focus on text classification and language modeling.
- Would be helpful to add more details regarding how the pertubed input is constructed for each linguistic capability. It is currently mentioned as the limitation (inefficient search methods), but i is unclear how inefficient since there is no explanation about it.
- Notation definition: more explicit in assigning $m_1$ and $m_2$, currently it only mentions $m_1$ as a reference model and I need to re-read in multiple place to infer what is $m_2$.
- I think the results for fairness capability is quite interesting. If there is space, might consider to put it in the main text.
- A relevant paper: https://arxiv.org/pdf/1711.02173.pdf

---

> ### Author Response · Authors · 2023-11-21
>
> Thank you Reviewer nq6A for a careful analysis of our paper! We are glad to know that you find our proposed framework useful and experiments well-executed. We address the concerns raised in your review below:
>
> **Are the four components really independent? Does transformation and constraint depend on each other?**
>
> We intentionally designed the independence of the four components from an implementation perspective by defining them separately in the code. This intentional design choice enables researchers and practitioners to define new linguistic capabilities tailored to their specific tasks in a modular fashion. We do agree that while defining a linguistic-capability one may implicitly weigh the interdependence of these components. For example, practitioners may choose not to impose specific constraints with particular transformations, or vice versa.
>
> **Puzzled with results of BERT-Tiny. Does this mean that the predictions of both models are quite different (hence poor agreement), but within the individual model the predictions do not change when it is given original and perturbed samples?**
>
> Precisely! We find that while larger target models (e.g., BERT-Base) disagreed with BERT-Tiny (as reference model) on individual samples (lower agreement), they still shared invariances i.e., the larger target models didn’t have substantial differences in their predictions on original and perturbed samples.
>
> **Any interesting findings on the model agreement on original input vs. perturbed input?**
>
> We didn’t observe any atypical trends on the model agreement on original input (IID-Agreement) and perturbed input (OOD-Agreement). We found them to mostly follow similar trends across many different settings. Along similar lines, a recent work in computer vision has also observed that IID-agreement and OOD-agreement are highly correlated with each other. [1]
>
> **Suggestions for presentation improvements**
>
> *Mention the tasks being used in the experiments at the beginning of the paper:* Thanks for the suggestion, we added a pointer to the tasks being used in experiments in the concluding paragraph of the Introduction.
>
> *Unclear how inefficient is perturbation generation:* The major computation costs for SCoPE can be attributed to the process of generating invariant perturbations w.r.t the reference model. Importantly, this is a one-time process (per reference model) and we can reuse the perturbations generated for a particular reference model for comparison with different target models. This cost varies depending on the linguistic capability. For instance, generating perturbations for Synonym-Invariance takes $\approx 1.2$ seconds per sample, for BERT-Base as a reference model. We have added a few results on the compute costs for perturbation generation in the supplementary material Sec. K. Please let us know If there are other specific details regarding perturbed input generation we can further clarify.
>
> *results for fairness capability is quite interesting. If there is space, might consider to put it in the main text:* We are glad to note that you find the fairness capability results quite interesting. Unfortunately, space constraints prevent us from providing a detailed discussion on the fairness capability results in the main paper. However, should our work be accepted, we plan to incorporate these findings into the main paper, as we would have access to an additional page.
>
> ---
>
> [1] Baek, Christina, et al. "Agreement-on-the-line: Predicting the performance of neural networks under distribution shift." Advances in Neural Information Processing Systems 35 (2022)

---

### Official Review · Reviewer_xMdc · 2023-10-31

**Soundness:** 2 fair
**Presentation:** 2 fair
**Contribution:** 1 poor
**Rating:** 3
**Confidence:** 4

**Summary:**

The paper proposes invariance-based metrics for capturing similarities between NLP models. Here, invariance-based metrics are contrasted with agreement-based metrics. To compute the proposed invariance-based metrics, one chooses a reference model (Ref) and constructs a dataset of (x, x’) invariant pairs. Here, x is the “base”, and x’ is the “perturbed” example, which is chosen as a perturbation of x that yields minimal change to Ref outputs. Two perturbation mechanisms that yield x’ candidates are considered, namely inserting typos and replacing words with synonyms. Given the (x, x’) pairs from Ref and a Target model (Tgt), one computes the similarity between how perturbing x to x’ changes Tgt behavior. This is contrasted to OOD agreement between Ref and Tgt, which would involve directly comparing the models’ predictions Ref(x’) and Tgt(x’) to each other. Instead, in the proposed invariance-based metrics Tgt(x’) is compared to Tgt(x). The paper proposes a hard and a soft metric, that are called Hard- SCoPE and Soft-SCoPE respectively.

The key message of the paper is that Soft-SCoPE sometimes does not correlate with OOD agreement. From this the paper infers that invariance-based metrics can offer complementary insights that agreement-based metrics may miss. For example, the paper shows how distilling BERT into Distill-BERT considerably decreases the Soft-SCoPE metric.

**Strengths:**

The paper aspires to address an important challenge of quantifying similarities and differences between different language models.

**Weaknesses:**

In my opinion, the paper fails to present a clear motivation and justification for the proposed metrics. For an auxiliary metric to be of interest to a practitioner, it needs to be predictive of what the practitioner cares about. In particular, the abstract of the paper refers to learning about “differences in model performance in the wild” as the motivation. I did not find an explanation of how measuring the proposed invariance-based metrics for pairs of models can help us get a better idea of the models’ “performance in the wild”. The main justification for the proposed metrics is that they don’t correlate with OOD agreement. But one can’t justify a new metric just by saying it’s different from existing ones - it is a logical fallacy.

I also found the paper rather difficult to follow. Here are some concerns as bullet items:
- Is a notation for the source of perturbed candidates x’ missing from Equation 1?
- A table showing examples of (x, x’) pairs is a must for a paper like this.
- Many complex sentences and reasoning chains make the paper hard to follow. Example (page 5): “Thus, measuring behavioral shared invariances amounts to quantifying the extent to which a target model is invariant on perturbations that do not cause any change in the reference model’s behavior (ref Eq. 1).”

**Questions:**

Why would a practitioner trust your metrics more than OOD-agreement to predict e.g. how much BERT distillation hurts the model’s robustness to typos?

---

> ### Author Response · Authors · 2023-11-21
>
> Thank you Reviewer xMdc for your thoughtful comments. We address the concerns raised in your review below:
>
> **Why do shared invariances tell us about performance in the wild?; Motivation of shared-invariances beyond OOD agreement**
>
> Being invariant to perturbations in data that are not intrinsically relevant to the task labels is helpful for models to generalize beyond their training distribution [1, 2]. Therefore, comparing the invariances shared between the two models can shed light on how similarly they will generalize on data that deviates from the training distribution. This is particularly interesting when considering models with distinct design choices (e.g., distillation, model size), as low values of shared invariance signify that the design choice renders the model susceptible to perturbations that it was originally robust to. In this work, we investigate three distinct perturbations – synonym-based word substitution, inducing misspellings, and protected category-based word substitution (fairness), that preserve the underlying sentiment of a sentence for a sentiment classification task. Through our experiments, we uncover several insights such as – distilling BERT makes it more susceptible to perturbations along certain linguistic capabilities (Typo-Invariance) than others, larger models encode a more diverse set of invariances compared to comparably accurate smaller models, etc.
>
> We would also like to clarify that we introduce novel measures of shared invariances based on the motivations described above. If existing metrics were able to capture the degree of shared invariance, then there would be no need to introduce new metrics. This is why we tested OOD-agreement, since it is one of the popular metrics to quantify the behavioral similarity of two models. To quantify the relationship between invariance- and agreement-based metrics, we conducted a correlation analysis between OOD-agreement and Soft-SCoPE which revealed poor correspondence in certain important cases. Therefore, if one wants to investigate the shared invariance between models, then using a metric like Soft-SCoPE is more desirable.
>
> **Why would a practitioner trust your metrics more than OOD-agreement to predict e.g. how much BERT distillation hurts the model’s robustness to typos?**
>
> OOD agreement can answer questions such as “Do BERT-Base and DistilBERT have the same behavior on a set of perturbed examples?”, while invariance-based metrics like SCoPE can answer questions such as “Does distilling BERT increase its susceptibility (or reduce invariances) to perturbations it was initially robust against?” The reason why OOD agreement on its own cannot answer the same question is that evaluating whether a target model is invariant to a perturbation requires determining whether it produces consistent outputs on both original and perturbed samples. In the special case of binary classification, the second-order agreement between IID and OOD agreement rates can be used for evaluating such invariances. We in fact explore this relationship in Sec. E.2 of the supplementary material in depth, and find that Hard-SCoPE (and consequently Soft-SCoPE) essentially measures this “agreement between the IID and OOD agreement rates” for a binary classification setup and hence evaluates the invariance of the target model directly.  Additionally, it becomes complicated to only use agreement rates to quantify invariances in a multi-classification setup. Therefore, SCoPE-based metrics prove to be more apt for assessing invariances encoded by a target model.
>
> **Is a notation for the source of perturbed candidates x’ missing from Equation 1?**
>
> We assume that by “source” here, the reviewer means initialization of $x’$. As discussed in Section 3.1, the linguistic capability $C$ perturbs $x$ to obtain the $x’$. Hence, the source of the $x’$ here is $x$. We subject the optimization procedure to disregard trivial solutions i.e., $x = x’$. We have updated Equation-1 in the paper to reflect the same more clearly.
>
> **A table showing examples of (x, x’) pairs is a must for a paper like this.**
>
> Thanks for the suggestion. We have added the same in the supplementary materials Sec. L.
>
> ---
>
> [1] Deng, Weijian, et al. "On the strong correlation between model invariance and generalization." Advances in Neural Information Processing Systems 35 (2022)
>
> [2] Izmailov, Pavel, et al. "On feature learning in the presence of spurious correlations." Advances in Neural Information Processing Systems 35 (2022)

---

### Official Review · Reviewer_cey2 · 2023-10-31

**Soundness:** 2 fair
**Presentation:** 1 poor
**Contribution:** 2 fair
**Rating:** 3
**Confidence:** 3

**Summary:**

This paper introduces SCoPE, a measure of invariance between language models (LMs). Unlike naive pointwise-agreement measures (that simply measure the fraction of inputs on which two models agree), SCoPE measures the fraction of robust (perturbation-robust) inputs on which models differ. As the perturbations are capability-defined, SCoPE is a measure targeted at a particular capability. Details follow.

Given two datasets $X$ and $X'$, a label set $Y$, a model $m\colon X \cup X' \to \mathbb{R}^n$ and a loss function $\ell \colon \mathbb{R}^n \times \mathbb{R}^n \to \mathbb{R}$, we define a _perturbation_ $p\colon X \to X'$ by mapping each $x \in X$ to $x'\in X'$ that minimizes $\ell(m(x), m(x'))$. [Note: In the paper it there is a linguistic capability perturbing the inputs, though I am unclear on how this is captured formally by the definition and so cannot include it in my summary.] Moving forward, we will denote by $y_m(x)$ the label of $x$ as predicted by $m$, that is, $y_m(x) = \mathrm{argmax}_{k \in [n]}m(x)_k$.

Given an input perturbation $x \to x'$ and two models $m_1$ and $m_2$, the Hard-SCoPE distance from $m_1$ to $m_2$ is defined by $\Pr_{x \in X}[y_{m_2}(x) = y_{m_2}(p(x)) \vert y_{m_1}(x) = y_{m_1}(p(x))]$. In plain words, we can say that Hard-SCoPE measures the degree to which inputs on which $m_1$ is perturbation-invariant, are also perturbation-invariant for $m_2$.

A variant of Hard-SCoPE, called Soft-SCoPE, is defined to be $f(m_1, m_2) \cdot \mathrm{Hard-SCoPE}$. The authors define $f(m_1, m_2) = \mathrm{decay}(\mathrm(dist)(\Delta \vec{m}_1, \Delta \vec{m}_2))$, however $\Delta\vec{m}$ is not defined and so I do not know how Soft-SCoPE is defined.

[Note: I did not understand the formal notation in the paper, so applied my own here to the best of my ability.]

The authors evaluate the utility of their measures by conducting various experiments. The perturbations are either synonym-based ("A man laughs out loud" $\to$ "A man laughs out loudly") or typo-based (a permutation of all but the first and last characters of a single word). Results are:

**BERT finetuned on sentiment analysis (Section 4)**
1. Between BERT (as the reference model $m_1$) and DistilBERT (as the target model $m_2$), SCoPE measures are higher for synonym-perturbations vs. typo-perturbations.
2. When comparing BERT-Tiny (as reference model) with synonym-perturbations to five BERT variants of different sizes, Hard-SCoPE _increases_ with model size, and Soft-SCope stays roughly the same. This is despite the fact that pointwise agreement on the perturbed dataset ("OOD Agreement") correlates with model size, i.e., follows the opposite trend of Hard-SCoPE.
3. When repeating the previous experiment with BERT-Base as a reference model, Hard-SCoPE and Soft-SCoPE now _do correlate_ with model size.

**GPT-2 vs. InstructGPT (Section 5)**
In the following, GPT-2 is used as the reference model against variants of InstructGPT
1. `text-davinci-001` has higher Hard and Soft-SCoPE than its smaller variants `text-ada-001`, `text-curie-001` and `text-babbage-001`.
2. `text-davinci-003` has higher Hard and Soft-SCoPE than `text-davinci-001` and `text-davinci-002`.

**Strengths:**

- The idea of measuring similarity of models by evaluating their agreement on perturbation-invariant inputs is, to my knowledge, novel.
- As opposed to new benchmarks or tasks, this measure can be computed based only on outputs (or logits) of the model, and is generic enough that it can be instantiated on any dataset (and its perturbation).
- The choice of models for experimentation provides an appealing setting, namely, understanding the effect of model size on robustness to perturbations.

**Weaknesses:**

### Significant Issues with SCoPE definition
There are numerous issues with the presentation of the definition of Hard-SCoPE and Soft-SCoPE. Considering that the new measure is the main contribution made in this paper, this is a significant weakness of this work. I could not attempt to infer the definition from its implementation, since the submission was not supplemented by code.

1. The notation $\Delta \vec{m}$ is never formally defined. Consequently, it is impossible to understand the definition of Soft-SCoPE.
2. In Hard-SCoPE and Soft-SCoPE, the expectation is said to be taken over $(x, x') \in (X, X')$. The domain $(X, X')$ is not defined. My first interpretation was that this was a cartesian product $X \times X'$, which means that $x, x'$ are taken iid--but this does not seem to align with the textual description of SCoPE. See my summary for an alternative way to denote taking $x$ and its corresponding perturbation.
3. In general, I do not follow (or do not agree with) the definition of a perturbation in this paper. For example, in page 4 you define $X'$ to be the image of $X$ under $C(\cdot, m)$. But this does not retain the mapping of inputs $x$ to their perturbations $C(x, m)$. Furthermore, $C$ is not well-defined, since the domain of the argmin is not specified. If we take the domain to be $X$, and read the follwing paragram in which you say that you take the loss function to be $\ell_1$-norm difference, then we have that $C$ ends up being the identity function (assuming the loss has no non-trivial zeroes).
4. The expectation of an indicator random variable is simply the probability over this event. It would be significantly more readable to use probabilities in the notation, as I did in may summary.
5. In Soft-SCoPE, since the attenuating factor $\mathrm{decay}(\mathrm{dist}(\cdot, \cdot))$ seems to be constant with respect to the random variables, it can be taken out of the expectation. Therefore, Soft-SCoPE is just Hard-SCoPE multiplied by a factor that depends on $m_1$ and $m_2$ (in an undefined way, see item 1 above).
6. In page 6 it is written that "Hard-SCoPE can only take binary values i.e., 0 or 1". My understanding of this sentence is that the Hard-SCoPE of two models can be either 0, or 1. But this is not consistent with the remainder of the text.

### Experiments lack depth
The goal of the experiments should be to argue for the utility of the novel measure introduced in this paper. This can be done, for example, by using this measure to obtain novel and interesting insight, or by dissecting the measure with fine-grained experiments.

In section 3, the authors compare the measure to naive pointwise measure (IID/OOD agreement). The main body of experiments (Sections 4 and 5) are conducted by taking different pairs of models and measuring Soft-SCoPE, Hard-SCoPE, and pointwise agreement. The experiments are then displayed with bar charts, and there is a textual description of the results. I found the results to be fairly superficial: the correlations are either expected (BERT-Base SCoPE correlates with model size), or uninterpretable (why is this not the case for BERT-Tiny?). When deeper conclusions are reached for, it is at times done in a speculative manner ("we hypothesize").

I unfortunately do not have concrete suggestions for a more refined experimental setup that will give more insightful results, in part because I could not understand the definition of the measure itself.

### Choice of "capabilities"
In the body of the paper, the perturbations are based on synonyms or permutations of characters in a word. I found the usage of the term "capability" to be confusing in this context. In NLP, I think of a capability as a task, such as sentiment analysis or summarization. Are these perturbations somehow related to capabilities as they are more generally used?

Additionally, were the choices of these pertubations arbitrary, or is there something about them that makes the particularly appropriate for exploring the newly-defined SCoPE measure? It seems to me that the results in Sections 4 and 5 could have been reported out all the same with other perturbations---this speaks to the genericness of the results (see previous weakness).

### Figure 5
I could not understand the diagram in the left part of Figure 5, despite making several attempts on different days. The diagram on the right would be clearer if presented by a table, as there are onalya three data points.

### Additional / minor issues
- Commas before / after "i.e." are inconsistent throughout the paper. There are different style guides for American vs. British English regarding this matter, but please pick a dialect and stick to it.
- Page 3: What is the argmin over? Presumably, over $x' \in X'$, but then $X'$ uses $C$ in its definition in the following page!
- Page 4: "and constraint modifications of words that are stopwords". Did you mean "constrain" (a verb) rather than "constraint" (a noun)?
- Page 4: There is a grammatical issue with the sentence starting with "We perform experiments along one..."
- Page 4: What is the argmax over? Presumably, over $k \in [c]$. It would be more readable to make this equation centered rather than inline.
- Page 4: The definition of $X'$ should use set builder notation: replace the $\forall$ with a colon or a vertical line.
- Page 5: In the definition of the acronaym you write **SH**ared-**C**-apabilities... But the **H** does not appear in the acronym. Furthermore, there is no need to hyphenate the definition.
- Page 5: "(ref Eq. 1)". Is the "ref" a typo? There is no such common shorthand, to my knowledge. Use "see" instead.
- Page 6: Use \mathrm{} for operators such as "decay" and "dist".
- Page 6: I suggest using the term "monotonically decreasing" rather than "has a downward slope".
- Page 7: In the first paragraph, there are several issues with citations and parentheses. There should probably not be sequences of parentheses such as "))" or ")(".

**Questions:**

I welcome responses and answers to any of the questions raised in the previous sections of my review. Besides these, I have two outstanding questions.

### What is the cost of computing SCoPE?
I could not find the computational costs reported in the paper or in the appendices. Since this paper proposes a new measure, it would be useful for the reader to know how expensive it is to compute. In particular, for the experiments in Section 5, the authors evaluate black-box models by taking random samples from their output; what is the monetary barrier to reproducing these experiments? Is there a compute-accuracy tradeoff inherent to this method?

### What is behavioral about the invariances?
The paper makes extensive references to "behavioral invariances". What is "behavioral" about these invariances? Is this term used in other parts of the literature, to distinguish thes from other kinds of invariances? "Behavior" alludes to some "cognitive" or grounded aspect of the model, or at the very least to its semantics, while typo-invariance is clearly a syntactic phenomenon. I think that using just "invariance" would be clearer.

---

> ### Author Response · Authors · 2023-11-21
> **Response to Reviewer cey2 (Part-1)**
>
> Thank you for reviewing our paper, Reviewer cey2. We address all the concerns you raised in your review below.
>
> **Issues with notations**
>
> *Notation delta_m1 is never formally defined:* $\Delta \vec{m}^{\}$ is the change in a model – $m$’s, output probability distribution in the presence of a perturbation ($x \rightarrow x’$), i.e., $\Delta \vec{m}^{\} = m(x')-m(x)$. Thus, Soft-SCoPE between two models Soft-SCoPE ($m_2$ | $m_1$) is high if the change (or effect of the perturbation) in two models ($\Delta \vec{m_1}^{\}$,$\Delta \vec{m_2}^{\}$) is similar (in both direction and magnitude) and low otherwise. We've improved the description for the same on Page-5/6.
>
> *Notation suggestions for perturbations and metrics:* Thanks for the suggestion regarding the alternative way of denoting notations. We have incorporated those suggestions and modified some of the notation definitions accordingly. In Hard-SCoPE and Soft-SCoPE the expectation is now over $x \in X$ and we denote the corresponding perturbed samples $x’$ in each equation by $C(x; m)$, where $m$ is the reference model. Additionally, while generating $x’$, we disregard the trivial solution i.e., $x = x’$, and optimize for a perturbation that minimizes the objective function $\mathcal{L}(m(x),m(x'))$.
>
> *Hard-SCoPE of two models can be either 0, or 1:* Thank you for the pointer. Your understanding is correct, we meant that the Hard-SCoPE of two models for a particular base-perturbed pair can either be 0 or 1. We have added clarifications on Page-6.
>
> **Experiments lack depth; correlations are either expected**
>
> We perform extensive experiments demonstrating the utility of the proposed measure across many different settings – different tasks (text classification and language modeling), datasets (SST2 and AG-News), linguistic capabilities (Synonym-Invariance, Typo-Invariance, Fairness), models (BERT-, InstructGPT-family), etc. Please let us know if you have are other specific suggestions for detailed experiments, and we would happy to follow up with them during the discussion period.
>
> **Choice of "capabilities"**
>
> *Relationship with perturbation:* As discussed in the Introduction and Related Works, we follow prior works [1, 2] in defining linguistic capabilities. A linguistic capability evaluates a model’s competence on a particular aspect of knowledge and understanding required to solve an NLP task by validating its input-output behavior under the corresponding scenario. Thus, these linguistic capabilities (such as invariance to synonyms, or variance to negation) are required by NLP models to perform a particular task (sentiment classification) well. One way to evaluate a model’s linguistic capabilities is via perturbations i.e., perturbing the input along the concept encoded by a linguistic capability (e.g., replacing words with synonyms, negating sentences) and evaluating whether the model’s prediction on the perturbed input aligns with human judgment. Thus, the perturbation generation rules are closely defined by the linguistic capability of interest.
>
> *Specific capabilities:* We focus on the particular linguistic capabilities i.e., Synonym-Invariance, Typo-Invariance, and Fairness as there is rich literature studying them in NLP, albeit from an adversarial robustness perspective as they concern the reliability of many real-world systems, such as spam detection, toxicity classification, and fake news detection [3, 4, 5]. We would like to emphasize that the primary goal of this work is to a) introduce a general framework for comparing shared invariances along a linguistic capability and b) proposing novel measures of behavioral shared invariance. The list of linguistic capabilities can be easily expanded by researchers and practitioners in the future based on the specifications of their research questions and tasks as they would only need to define the specific transformation and constraints.
>
> **Confusion regarding Figure 5:**
>
> In Figure 5 (left), we aim to explore the effect of the size of different black-box GPT-3 models when compared with GPT-2 as the reference model. The x-axis represents different metrics, while each different color corresponds to a distinct GPT-3 target model. We keep the styling template similar to the one in Figure 4, where we also compare multiple target models across multiple metrics. Please let us know if there’s anything specific that’s unclear, we would be happy to clarify further.
>
> **Additional / minor issues**
>
> Thank you for these suggestions, we have incorporated them in the paper.

---

> ### Author Response · Authors · 2023-11-21
> **Response to Reviewer cey2 (Part-2)**
>
> **What is the cost of computing SCoPE?**
>
> The major computation costs for SCoPE can be attributed to the process of generating invariant perturbations with respect to the reference model. Importantly, this is a one-time process (per reference model) and we can reuse the perturbations generated for a particular reference model for comparison with different target models. This cost varies depending on the linguistic capability, and we have added a few results on the same in the supplementary material Sec. K. There is no substantial additional compute overhead on computing the SCoPE-based metrics, once perturbations are generated.
>
> Regarding the compute-accuracy tradeoff:
> In Section H of the supplementary materials, we discuss the reliability of our proposed metrics as the number of base samples (X) decreases. Specifically, we report the mean metric values and the 95% confidence interval of this estimate computed over 100 trials for many values of the base-sample count. We find that our metrics provide tight confidence intervals for as low as 50 base samples. For experiments in Section-5, since the reference model was fixed as GPT-2, the costs were primarily due to running each GPT-3.5 model on 100 base-perturbed sample pairs. At the time of conducting the experiments, it cost us $\approx 55$\$ to generate all the results in Section-5.
>
> **What is behavioral about the invariances?**
>
> The invariances shared between models can be evaluated either at the output level, which we refer to as the behavior of the model, or at the representation level, examining whether the model maintains very similar representations (at some intermediate layer) for perturbed and original samples. In the related works section, we discuss works [6] that compute representational shared-invariances between two computer vision models, and elaborate why their setups may not be directly applicable to natural language processing.
>
> ---
>
> [1] Ribeiro, Marco Tulio et al.  Beyond accuracy: Behavioral testing of NLP models with CheckList. In Proceedings of the 58th Annual Meeting of the Association for Computational Linguistics,
>
> [2] La Malfa, Emanuele, and Marta Kwiatkowska. The king is naked: on the notion of robustness for natural language processing. Proceedings of the AAAI Conference on Artificial Intelligence.
>
> [3] Lee, H. and Ng, A. Y. Spam deobfuscation using a hidden markov model. In CEAS, 2005.
>
> [4] Pruthi, D. et al., Combating adversarial misspellings with robust word recognition. In Proceedings of the 57th Annual Meeting of the Association for Computational Linguistics.
>
> [5] Ren, S. et al., Generating natural language adversarial examples through probability weighted word saliency. In Proceedings of the 57th Annual Meeting of the Association for Computational Linguistics
>
> [6] Nanda, Vedant, et al. "Measuring Representational Robustness of Neural Networks Through Shared Invariances." International Conference on Machine Learning. PMLR, 2022.

---

> > ### Comment · Reviewer_cey2 · 2023-11-21
> >
> > Thank you for your response. Some follow-ups:
> >
> > ### Issues with notation
> > 1. Thank you for defining $\Delta \vec{m}$. Unfortunately, the definition provided ($\Delta \vec{m} = m(x) - m(x')$) is not clear to me, as it is not clear which $x$ and $x'$ are taken in the right hand side. Is it the expected difference under a random $x$ and its perturbed counterpart $x'$?
> > 2. The change makes the expectation well-defined, and is clearer to me now---thanks! However, the change needs to be made throughout pages 4-6, as the remainder of the text still uses the old ($x \to x'$) notation, while the equations themselves use $C(x;m)$.
> >
> > Concerns 3-5 remain. Thank you for the clarification and fix regarding concern 6.
> >
> > ### Cost of computing SCoPE
> > Your comment addressed my concern. I suggest incorporating this useful explanation into the paper.
> >
> > ### Choice of "capabilities"
> > CheckList lists various capabilities in section 2.1, and indeed their _Robustness_ corresponds to the invariances considered in this paper. To avoid a confusion such as the one in my initial review, it could be helpful to explain this connection.
> >
> > ### Figure 5 (actually Figure 1, my bad)
> > My concerns were regarding Figure 1, not Figure 5. I apologize for the mistake in the original review; I will not edit it so that our discussion is clearer.
> >
> > # Score update
> > In light of my concerns that were addressed, as well as those that remain, I have increased the soundness and overall scores of the paper.

---

> > > ### Author Response · Authors · 2023-11-22
> > > **Thank you for the follow-up comments!**
> > >
> > > **Issue with notation**
> > >
> > > C1. $\Delta \vec{m}^{\} = m(x')-m(x)$ is the change in model $m$’s output for a specific base-perturbed sample pair i.e., a particular base sample $x \in X$ and its perturbed counterpart $x’ = C(x; m_{ref})$, where $m_{ref}$ is the reference model which may not be the same as $m$. We have revised Equation 3 as the decay(dist($\Delta \vec{m_1}^{\}$, $\Delta \vec{m_2}^{\}$)) term depends on $x \in X$.
> > >
> > > C2. We have updated the paper to explicitly clarify that $x’ = C(x; m_{ref})$ at places where we use the notation $x \rightarrow x’$. Note $m_{ref}$ refers to the reference model and is replaced appropriately in the text depending on the context.
> > >
> > > C3. As referred to in our previous rebuttal, while minimizing the $l_{1}$-norm difference we disregard trivial solutions that correspond to $x = x’$. Thus, we aim to find the perturbation that has minimal impact on the reference model’s behavior. We have updated Equation 1 to reflect the same.
> > >
> > > C4. As alluded to in C1, the Soft-SCoPE formulation incorporates the decay(dist(., .)) term within the expectation. Hence, we use the expectation notations for the sake of consistency.
> > >
> > > C5. Please refer to C1. and C4.
> > >
> > > **Include cost of computing SCoPE in main paper**
> > >
> > > Thanks for the suggestion! We have added a pointer regarding the costs incurred in running the GPT-3 experiments in Section-5 (Page 9). We have included a small discussion on the computational cost of search methods for generating perturbations along different linguistic capabilities in Sec. 3.1 (Page 3).
> > >
> > > **Choice of “capabilities**
> > >
> > > We agree that explaining this connection with CheckList more clearly could help avoid confusion regarding the choice of linguistic-capabilities in our experiments. Given the limitations on space in the current version of the paper, a detailed discussion on the same is not feasible at this time. We plan to address this concern by incorporating a thorough explanation in Section 3.2 of the camera-ready version of the paper, should it be accepted.
> > >
> > > **Confusion regarding Figure 1**
> > >
> > > *Could not understand the diagram in the left part of Figure 1:*
> > > Consider a model – $m$, trained on a binary classification task where the labels correspond to two classes – Class-A and Class-B.
> > > For a particular input – $x$, we can visualize the output softmax probabilities of $m$ i.e., m(x) = [0.8, 0.2] via vectors on a 2-D plane where the y-axis corresponds to the Class-A probability score (0.8 here) and x-axis corresponds to the Class-B probability score (0.2 here).
> > > In Figure 1 (left), we visualize the output softmax probability vectors for three models $m_1$, $m_2$, and $m_3$ for a pair of original-perturbed sample ($x$, $x’$), where $x’ = C(x; m_1)$. Note, in this example $m_1$ is the reference model, while  $m_2$ and $m_3$ are target models.
> > > We observe that both $m_2$ and $m_3$ remain consistent in their predictions i.e., $y_{m_{2}}(x)$ = $y_{m_{2}}(x’)$ and $y_{m_{3}}(x)$ = $y_{m_{3}}(x’)$, as they predict Class-B on both original and perturbed samples. Thus, the Hard-SCoPE values – Hard-SCoPE ($m_2 \mid m_1$) and Hard-SCoPE ($m_3 \mid m_1$) is 1. However, we note that the effect of the perturbation (in both direction and magnitude), as measured by $\Delta \vec{m}^{\} = m(x')-m(x)$ , is much more aligned for one pair of target-reference model i.e., $\Delta \vec{m_3}^{\}$ and $\Delta \vec{m_1}^{\}$ than the other pair $\Delta \vec{m_2}^{\}$ and $\Delta \vec{m_1}^{\}$.
> > > Thus, the reliance on (argmax) predictions to quantify shared invariances by Hard-SCoPE obscures the finer differences between the effect of the perturbation. As motivated by this example, we propose a softer measure of shared-invariances – Soft-SCoPE that accounts for such differences and assigns higher values when the effect of the perturbation in both models is similar (e.g., 1 for $\Delta \vec{m_3}^{\}$ and $\Delta \vec{m_1}^{\}$) and low values otherwise (e.g., 0.6 for $\Delta \vec{m_2}^{\}$ and $\Delta \vec{m_1}^{\}$).
> > >
> > > *Diagram on the right would be clearer if presented by a table, as there are only three data points:*
> > > We keep Figure 1 (right) as a figure as the key focus here is not on the three data points – $\Delta \vec{m_1}^{\}$, $\Delta \vec{m_2}^{\}$ and $\Delta \vec{m_1}^{\}$, but rather on the smooth variation of the Soft-SCoPE score as the effect of perturbation in the target model becomes less/more aligned with that in the reference model.

---

### Official Review · Reviewer_yXpb · 2023-11-01

**Soundness:** 3 good
**Presentation:** 3 good
**Contribution:** 3 good
**Rating:** 8
**Confidence:** 3

**Summary:**

The paper studies invariances in different semantic or linguistic features across LLM representations. The authors propose a novel metric study for evaluating the invariance in LLMs. Paper includes extensive experiments across metrics and models. It would be a good fit for the conference and informative contribution.

**Strengths:**

The paper includes extensive comparison across existing metrics and prominent LLMs to help understand their capabilities.

**Weaknesses:**

Description of experimental methodology needs to be clarified.

**Questions:**

IID not defined.

---

> ### Author Response · Authors · 2023-11-21
>
> Thank you for your review Reviewer yXpb.
>
> We have added descriptions for the previously undefined acronyms, namely, IID (Independent and identically distributed) and OOD (Out-of-Distribution) data in the paper. We have also included additional experiment details related to different NLP tasks in the concluding paragraph of the Introduction. Please let us know if there are any other specific details that we can further clarify.

---

### Meta-Review · Area_Chair_hpmF · 2023-12-03

**Metareview:**

The paper introduce a measure of invariance between language models that quantifies the fraction of perturbation-robust inputs on which models differ. The measure is assessed on a set of linguistically motivated perturbations, across various models.

The reviewers and I agree that this approach is novel and potentially useful to research on language model understanding. However, two major concerns make the paper, in my opinion and in that of two reviewers, not strong enough in its current version to be published at ICLR.

First, even after the revision, it is extremely difficult to fully understand the definition of SCoPE, the introduced measure, from the paper (as a matter of fact, I found reviewer cey2's explanation somewhat more helpful than the one in the original version of the paper).

Second, the experiments do not really provide insights on why we should be using the measure and what we gain from it.

**Justification For Why Not Higher Score:**

While the paper presents a promising idea, it requires further clarification of the main idea and more insightful experiments.

**Justification For Why Not Lower Score:**

N/A

---

### Decision · Program_Chairs · 2024-01-16

Reject